# A newly detected bias in self-evaluation

**Guillaume Deffuant**[1,2]*, **Thibaut Roubin**[1], **Armelle Nugier**[2], **Serge Guimond**[2]

**1** Université Clermont Auvergne, INRAE LISC, Aubière, France, **2** Université Clermont Auvergne, LAPSCO, Clermont-Ferrand, France

* guillaume.deffuant@irstea.fr

## Abstract

The widely observed positive bias on self-evaluation is mainly explained by the self-enhancement motivation which minimizes negative feedbacks and emphasizes positive ones. Recent agent based simulations suggest that a positive bias also emerges if the sensitivity to feedbacks decreases when the self-evaluation increases. This paper proposes a simple mathematical model in which these different biases are integrated. Moreover, it describes an experiment (N = 1509) confirming that the sensitivity to feedbacks tends to decrease when self-evaluation increases and that a directly related positive bias is detected.

**Data Availability Statement:** All data and programs are publicly available at: https://github.com/guillaumeDeffuant/sensitivityBias.

**Funding:** In complement to our earlier statement that "the research has been partly supported by the French National Research Agency through the

## Introduction

*People overrate themselves. On average, people say that they are "above average" in skill, over-estimate the likelihood that they will engage in desirable behaviors and achieve favorable outcomes, furnish overly optimistic estimates of when they will complete future projects, and reach judgements with too much confidence.*

This quotation from [1] is part of a review of numerous evidences of a positive bias in self-evaluation. This review reports in particular evidence of overoptimism or overconfidence in judgement and predictions, for instance about the duration of a romantic relationship [2] or the ability to complete a task [3] or about forecasting events in general [4–7].

A general explanation of this positive bias is the self-enhancement motive, which is the drive to convince ourselves, and any significant others in the vicinity, that we are intrinsically meritorious persons: worthwhile, attractive, competent, lovable, and moral [8].

Self-enhancement manifests itself in a variety of processes [8]. For instance, when people describe events in which they were involved, they tend to attribute positive outcomes to themselves, but negative outcomes to others or to circumstances, thus making it possible to claim credit for successes and to disclaim responsibility for failures [9, 10]. People also tend to remember their strengths better than their weaknesses [11, 12]. Where a threat to ego cannot be easily ignored, people will spend time and energy trying to refute it. A familiar example is the student who unthinkingly accepts a success to an examination but mindfully searches for reasons to reject a failure [13]. Note that, in order to achieve self-deception, these processes should be at least partially unconscious. Moreover, it seems that they activate various neural patterns, depending on the type of threat to the ego [14].

grant number ANR-18-ORAR-0003-01 (ToRealSim project)", we confirm again that there was no additional external funding received for this study. Moreover, the funder had no role in study design, data collection and analysis, decision to publish, or preparation of the manuscript.

**Competing interests:** The authors have declared that no competing interests exist.

If self-enhancement seems to be a generally dominant motive, in some contexts the motives of self-assessment, self-confirmation or self-improving may prevail [15, 16]. Moreover, in some contexts, people tend to evaluate themselves consistently worse than the average others [17]. In this respect, the experiment reported in [18] is an important example for this paper because our own experiment shares several features of its design. Surprisingly, the results of [18] show that the participants self-derogate instead of self-enhancing. Indeed, they give more influence to negative feedbacks than to positive ones, hence they show a negative bias. This experiment repeats sequences where the participants perform a task, then receive a feedback about their performance at the task and try to improve their performance at the next sequence. The feedback is actually defined by the experimenters and completely disconnected from the performance of the participants. The authors argue that because the participants are learning a task and try to improve their performance, negative feedbacks are more informative and thus are given more weight, which induces a negative bias.

In this paper, we consider a simple model of self-enhancement or self-derogation, leading to a positive or a negative bias in self-evaluation [18, 19]. The model represents an agent holding a self-evaluation and changing this self-evaluation when receiving feedback. The feedback is said positive when it is higher than the agent's self-evaluation and negative in the opposite case. The agent increases its self-evaluation when receiving a positive feedback and decreases it when receiving a negative feedback. Then, self-enhancement is the tendency of the agent to react more strongly to positive than to negative feedbacks. In this case, the agent increases its self-evaluation more strongly when receiving positive feedbacks than it decreases its self-evaluation when receiving negative feedbacks. By contrast, the agent shows self-derogation when it reacts more strongly to negative than to positive feedbacks. Self-enhancement causes a positive bias, namely a self-evaluation higher than the average of received feedbacks, while the self-derogation causes a negative bias, namely a self-evaluation lower than the average of received feedbacks. This model clarifies the distinction between the processes, self-enhancement or self-derogation, and their effect, a positive or a negative bias in self-evaluation.

More importantly in our work, an additional positive bias, generated by a completely different process, appears in this model. This bias is the main subject of this paper and, as far as we know, it is absent from the social-psychology literature. This additional bias appears when the sensitivity of the self-evaluation to the feedbacks (whether positive or negative) decreases when the self-evaluation increases, independently from self-enhancement, self-derogation or perfectly symmetric reactions to positive and negative feedbacks. Its cause is purely statistical and it has nothing to do with any motivation related to self. We call it bias from decreasing sensitivity to feedbacks or in short, bias from sensitivity.

We firstly observed this bias on a more complex agent based model [20]. Indeed, in this model, the agents have perfectly symmetric reactions to positive and negative feedbacks (no self-enhancement and no self-derogation), therefore the bias from sensitivity appears alone which makes it more easily observable. In other conditions, there is no means to disentangle it from the effect of self-enhancement or of self-derogation, without applying a specifically designed mathematical treatment. Moreover, understanding the process generating this bias required serious efforts [21] and it seems extremely unlikely that anyone would conceive this process without having detected the bias. It is therefore unsurprising that we did not find any paper about this bias in the social-psychology literature.

In this paper, our primary objective is precisely to detect this bias from sensitivity to feedbacks in humans after its observation in computer agents. More precisely, we aim at bringing experimental evidence supporting two main hypotheses:

1. People show an average decreasing sensitivity to feedbacks, when their self-evaluation increases;

2. This induces a specific positive bias in self-evaluation that is added to self-enhancement or self-derogation bias.

We report the results of an experiment (N = 1509) designed with this aim. The participants in this experiment perform a task once and then evaluate their performance at the task several times in reaction to evaluations (feedbacks) given by the experimenters. From data coupling self-evaluation and change of self-evaluation in reaction to a feedback, we compute a linear regression approximating the average sensitivity to feedbacks as a function of self-evaluation in different sets of participants. The slope of this linear regression is significantly negative especially when computed from the participants who believed that the feedbacks were real, showing that the sensitivity to feedbacks is decreasing. Moreover, we measure a significant average positive bias from this decreasing sensitivity to feedbacks, additional to the self-enhancement (or self-derogation) bias, especially in the set of participants who believed that the feedbacks were real. These results provide evidence supporting our hypotheses.

Our secondary objective is to observe how both sensitivity and self-enhancement biases are moderated by variables such as evaluation scale, gender and self-esteem. Of course, we did not find any publication about the effect of these variables on the sensitivity bias, since, as far as we know, the very existence of this bias has not been envisaged up to now. Hence this part of the paper can be seen as exploratory.

Surprisingly, we did not find any previous theoretical or experimental work about the effect of the evaluation scale on self-enhancement and our results about this effect can also be seen as exploratory. By contrast, there is a very abundant literature about the effects of gender and self-esteem on self-enhancement. Obviously, it is out of the scope of this paper to provide an exhaustive review of this literature and we limit our analysis to selected references that appear particularly relevant in the context of our experiment.

From this analysis, that we report in more details in the discussion, we conclude that:

- The literature points to an increase of self-enhancement with self-esteem. The main rationale behind this effect is that people with high self-esteem tend to be more motivated to protect or increase their self-esteem [22, 23] or high self-esteem tends to increase the self-deceptive mechanisms of self-enhancement [24].

- In the case of our experiment, the literature predicts that men will self-enhance more than women. Indeed, first, in our experiment, the task is related to a rather masculine subject and men tend to self-enhance more than women with respect to such subjects [25, 26]. Second, the evaluation is based on individual performance, closely related to the agency domain, in which men tend to self-enhance more than women [27, 28]. Finally the performance is not publicly disclosed, hence the self-enhancement is rather self-deceptive and men are more easily subject to this self-deception than women who engage more easily in impression management [29].

With respect to our secondary objective, our main finding is that the average sensitivity bias remains relatively stable when scale, gender and self-esteem vary (in the group of participants who believed in the feedbacks). This stability of the sensitivity bias is remarkable because the closely related self-enhancement bias varies very significantly, primarily with the scale.

Indeed, when the scale decreases with the evaluation, we find a significant self-enhancement bias as expected, but when the scale increases with the evaluation, we systematically find a significant self-derogation bias, like in the experiment of [18] (who used an increasing scale).

This effect, which, as far as we know, has not been reported in the literature, questions the explanation of the self-derogation by the motive to learn proposed in [18].

Moreover, in line with our analysis of the literature, the group of participants with high self-esteem and men tend to show a higher self-enhancement or a weaker self-derogation.

The next section presents our hypotheses in more details, the experimental setting and the method used for treating the results. The following section reports the results of the experiment. The final section proposes a discussion about them.

## Materials and methods

### Model and hypotheses

This section presents the model of an agent modifying its self-evaluation when receiving feedbacks. This is a simplified version of the agent model described in [20, 21]. It shows how a positive bias emerges from different series of feedbacks, if the sensitivity to feedback decreases when the self-evaluation increases. Then it extends the approach to a model that includes self-enhancement or self-derogation.

**General definition of the positive bias from decreasing sensitivity.** Consider an agent with self-evaluation $a_t$ at time $t$ when receiving feedback $f_t$ (i.e. an evaluation coming from an outside source). The main hypotheses from [20] are:

- the change of self-evaluation due to this feedback is proportional to the difference between the feedback and the self-evaluation;

- Moreover, the coefficient of proportionality decreases with $a_t$.

These hypotheses are thus expressed by Eq 1:

$$a_{t+1} - a_t = h(a_t)(f_t - a_t), \tag{1}$$

where $h(a_t)$ is a positive and decreasing function (after averaging possible random fluctuations) that we call "sensitivity to feedbacks". In the following, we assume that the sensitivity $h$ is derivable, thus its derivative $h'$ is negative: $h'(a_t) < 0$ for all $a_t$.

According to this model, for the same difference between feedbacks and self-evaluations (whether positive or negative), agents with a high self-evaluation are less influenced than agents with a low self-evaluation.

This model is a particular case of the model of interactions between two agents in [20, 21, 30]. Indeed, for any pair agent 1 and agent 2 in the population, the latter model assumes that the more agent 1 feels superior to agent 2, the lower the influence of agent 2 on agent 1 and the more agent 1 feels inferior to agent 2, the higher the influence of agent 2 on agent 1. Mathematically, the influence of agent 2 on agent 1 is a decreasing function $h(a_{11} - a_{12})$, where $a_{11}$ is agent 1's self-opinion and $a_{12}$ is the opinion of agent 1 about agent 2. Eq 1 is the same model when assuming that the opinion $a_{12}$ of agent 1 about agent 2 is constant.

In our experiment, agent 1 is the participant and agent 2 is the source of the feedback, which is described in more details later. We assume that the participants have no reason to change their opinion about the source of feedback over time. Hence in this perspective, postulating Eq 1 can be seen as neglecting the variations of the opinion of the participant about this source.

However, Eq 1 can also address cases where the feedback does not come directly from another agent but is a general evaluation from the environment, like a failure or a success. In this case, a possible justification is that agents with a high self-evaluation tend to be more confident and this makes them less prone to change their mind. The general hypothesis is that

people having a high self-evaluation are less easily influenced than people having a low self-evaluation.

The fact that the sensitivity to feedbacks $h$ is decreasing induces a general positive bias that we define mathematically now. Assume that the feedback is a random distribution of average $a_1$, which is also the initial self-evaluation. The first feedback is $f_1 = a_1 + \theta_1$, $\theta_1$ being randomly drawn from the distribution of average 0. The self evaluation after receiving this feedback is:

$$a_2 = a_1 + h(a_1)\theta_1. \tag{2}$$

Then, after the second feedback $f_2 = a_2 + \theta_2$, $\theta_2$ being randomly drawn from the distribution around $a_1$, the self-evaluation $a_3$ after receiving this feedback is:

$$a_3 = a_2 + h(a_2)(a_1 + \theta_2 - a_2) \tag{3}$$

Assuming that $\theta_1$ is small, the sensitivity $h(a_2)$ at $a_2$ can be approximated at the first order as:

$$h(a_2) = h(a_1) + h'(a_1)h(a_1)\theta_1. \tag{4}$$

Replacing $a_2$ by its value and $h(a_2)$ by this approximation yields:

$$a_3 = a_1 + h(a_1)\theta_1 + (h(a_1) + h'(a_1)h(a_1)\theta_1)(\theta_2 - h(a_1)\theta_1), \tag{5}$$

$$= a_1 + h(a_1)\theta_1 + h(a_1)(\theta_2 - h(a_1)\theta_1) + h'(a_1)h(a_1))(\theta_2\theta_1 - h(a_1)\theta_1^2). \tag{6}$$

Because we assume the averages of $\theta_1$ and of $\theta_2$ are 0, the average $\overline{a_3}$ of $a_3$ over all possible draws of $\theta_1$ and $\theta_2$ is:

$$\overline{a_3} = a_1 - h'(a_1)h^2(a_1)\overline{\theta_2^2}. \tag{7}$$

As we assume $h'(a_1) < 0$, we always have:

$$-h'(a_1)h^2(a_1)\overline{\theta_2^2} > 0. \tag{8}$$

This value defines the positive bias. The second evaluation $a_3$ is on average higher than the average feedback $a_1$ because of this bias.

This result extends to longer series of feedbacks [21]. The positive bias increases with the length of the series to an asymptotic value, which remains of the second order (in $\overline{\theta^2}$).

Our main aim is to check experimentally the existence of this bias. If we directly derive the experiment from the previous formulas, we face a hard problem: we need a huge number of random draws of feedbacks in order to get their average close to 0 and get a chance to detect the bias. To overcome this difficulty, we consider particular series of feedbacks in which the bias appears without averaging over many trials.

**Positive bias from decreasing sensitivity with alternating positive and negative feedbacks.** Let $f_t - a_t$ be the intensity of feedback $f_t$. We say that a feedback is positive when its intensity is positive and negative otherwise. We show now that the previous model generates a positive bias when receiving a series of feedbacks of opposite intensities. We consider the simple example of an agent receiving two consecutive feedbacks of opposite intensities $\pm\delta$.

Assume that the agent starts with self-evaluation $a_1$ and receives first the positive feedback $f_1 = a_1 + \delta$. Applying Eq 1, the self-evaluation of the agent becomes $a_2$:

$$a_2 = a_1 + h(a_1)\delta. \tag{9}$$

Then the agent receives the negative feedback $f_2 = a_2 - \delta$ and its self-evaluation $a_3$ becomes:

$$a_3 = a_2 - h(a_2)\delta. \tag{10}$$

The difference of self-evaluation between before and after receiving the couple of feedbacks is:

$$a_3 - a_1 = a_1 + h(a_1)\delta - h(a_2)\delta - a_1 = (h(a_1) - h(a_2))\delta. \tag{11}$$

As we assume that at any time $t$, $h(a_t) > 0$, we have $a_1 < a_2$ and, as $h$ is decreasing, we have: $h(a_1) - h(a_2) > 0$, hence $a_3 - a_1 > 0$.

If we invert the order of the feedbacks ($f_1 = a_1 - \delta$ and $f_2 = a_2 + \delta$), we have:

$$a_3 - a_1 = (h(a_2) - h(a_1))\delta. \tag{12}$$

Now $a_2 < a_1$, therefore again, because $h$ is decreasing $a_3 - a_1 > 0$.

Therefore, after receiving two feedbacks of opposite intensities, the self-evaluation tends to increase.

Developing $h(a_2)$ at the first order like previously, we can approximate the value of the bias:

$$h(a_2) \approx h(a_1) + h'(a_1)h(a_1)\delta, \text{ if } f_1 = a_1 + \delta; \tag{13}$$

$$h(a_2) \approx h(a_1) - h'(a_1)h(a_1)\delta, \text{ if } f_1 = a_1 - \delta. \tag{14}$$

Therefore, for both sequences of feedbacks we get:

$$S(a_1) = a_3 - a_1 \approx -h'(a_1)h(a_1)\delta^2. \tag{15}$$

This positive bias is thus expected to be of the second order of the intensity of the feedback, hence rather small. With a series of feedbacks of opposite intensities, the positive bias appears directly, without requiring to average on a large number of trials. In an experiment, the participants processing such a series of feedbacks of opposite intensities are expected to provide a noisy value of function $h(a)$ for each self-evaluation $a$ in the series. We expect to approximate the average value of $h(a)$ and the related bias when computing them from data collected on a reasonable number of participants. Up to now, we have assumed that the agent self-evaluates without self-enhancement, because in Eq 1, the sensitivity to the positive feedback $a_t + \delta$ and to the negative feedback $a_t - \delta$ are the same: $h(a_t)$. We now extend the model to the case where these functions are different.

**Positive bias from decreasing sensitivity with self-enhancement or self-derogation.** In the framework of this model, self-enhancement or self-derogation take place when the sensitivity $h_p(a_t)$ to positive and $h_n(a_t)$ to negative feedbacks are different:

$$a_{t+1} - a_t = h_p(a_t)\delta, \text{ if } f_t = a_t + \delta, \tag{16}$$

$$a_{t+1} - a_t = -h_n(a_t)\delta, \text{ if } f_t = a_t - \delta. \tag{17}$$

In the following, for sake of simplicity, we use self-enhancement in a general sense which includes self-derogation, considered as a negative self-enhancement. When induced by feedbacks of intensity $\pm\delta$, the bias of self-enhancement $E(a)$ at a given self-evaluation $a$ can be expressed as the difference between the reaction to the positive feedback $f_p = a + \delta$ and the

reaction to the negative feedback $f_n = a - \delta$:

$$E(a) = (h_p(a) - h_n(a))\delta. \tag{18}$$

Now, assume that the agent's self-evaluation is $a_1$ and that the agent receives a positive and then a negative feedback. Repeating the previous calculations, we get:

$$a_2 = a_1 + h_p(a_1)\delta, \tag{19}$$

$$a_3 = a_2 - h_n(a_2)\delta. \tag{20}$$

The total bias $B(a_1)$ from these successive feedbacks is:

$$B(a_1) = a_3 - a_1 \tag{21}$$

$$= (h_p(a_1) - h_n(a_1))\delta - h'_n(a_1)h_p(a_1)\delta^2. \tag{22}$$

We recognise the self-enhancement bias (Eq 18) in the first term and the bias from decreasing sensitivity (Eq 15) in the second term. For this sequence of feedbacks, the bias from decreasing sensitivity is thus:

$$S(a_1) = -h'_n(a_1)h_p(a_1)\delta^2. \tag{23}$$

This value is positive when $h'_n(a_1)$ is negative and we have:

$$B(a_1) = E(a_1) + S(a_1). \tag{24}$$

Moreover, if we have a series of 2 positive and 2 negative feedbacks in a random order (as it will be the case in the experiment), the average bias from decreasing sensitivity is:

$$S(a) = \frac{1}{4}\left(-h'_n(a)h_p(a) - h'_p(a)h_n(a) - h'_p(a)h_p(a) - h'_n(a)h_n(a)\right)\delta^2, \tag{25}$$

$$S(a) = -h'_m(a)h_m(a)\delta^2, \tag{26}$$

where $h_m$ is the average of $h_p$ and $h_n$: $h_m(a) = \frac{1}{2}(h_p(a) + h_n(a))$. In the following experiments, we approximate functions $h_n$ and $h_p$ with linear regressions from data collected on several participants. Then we evaluate the biases from self-enhancement and decreasing sensitivity using the above formulas.

## Experiment

The experiment design has been approved by the committee of ethics from Clermont Auvergne Université (reference number IRB00011540–2020-39). The participants live in France and were recruited online by a specialised company which verifies that they are not bots by checking an ID number. The participants are explicitly requested to give their consent by ticking a specific button that enables them to start the questionnaire.

The participants receive a series of 4 feedbacks, two positive, two negative, of same intensity in absolute value, starting from different self-evaluations. From these data, we compute linear approximations of the sensitivities to positive and negative feedbacks (functions $h_p$ and $h_n$ in the model) and then the different biases. We can thus reach our primary objective, which is to check the existence of a bias from decreasing sensitivity combined with the bias from enhancement, and our secondary objective, which is to observe how the biases are modulated by other variables (scale, gender, self-esteem).

**Overview of the experiment.** The experiment is schematically presented on Fig 1. The type of task and the evaluation by comparison with the performances of a large group, using a scale between 0 and 100, are similar to the ones used in [18]. The participants answer to an online questionnaire which includes the following steps:

- The participants are requested to assess the size of the coloured surface in the 3 different 2D images (see an example of image on the top left of Fig 1).

- The participants are told that the experimenters can compute exactly their error of surface assessment on these three images and can do the same for a large number of other people who already performed the task. Moreover, the participants are told that the experimenters gathered at random 6 different groups ($G_0$ to $G_5$) of errors from 100 people and that the error of the participant will be compared to these groups. This comparison provides an evaluation, between 1 and 100, of the participant with respect to the group. We tested two evaluation scales: rank and score which are described further.

- We assume that, initially, the participants have no idea of their self-evaluation at the task. Therefore, the participants are given the initial evaluation $f_0$ of their error with respect to $G_0$,

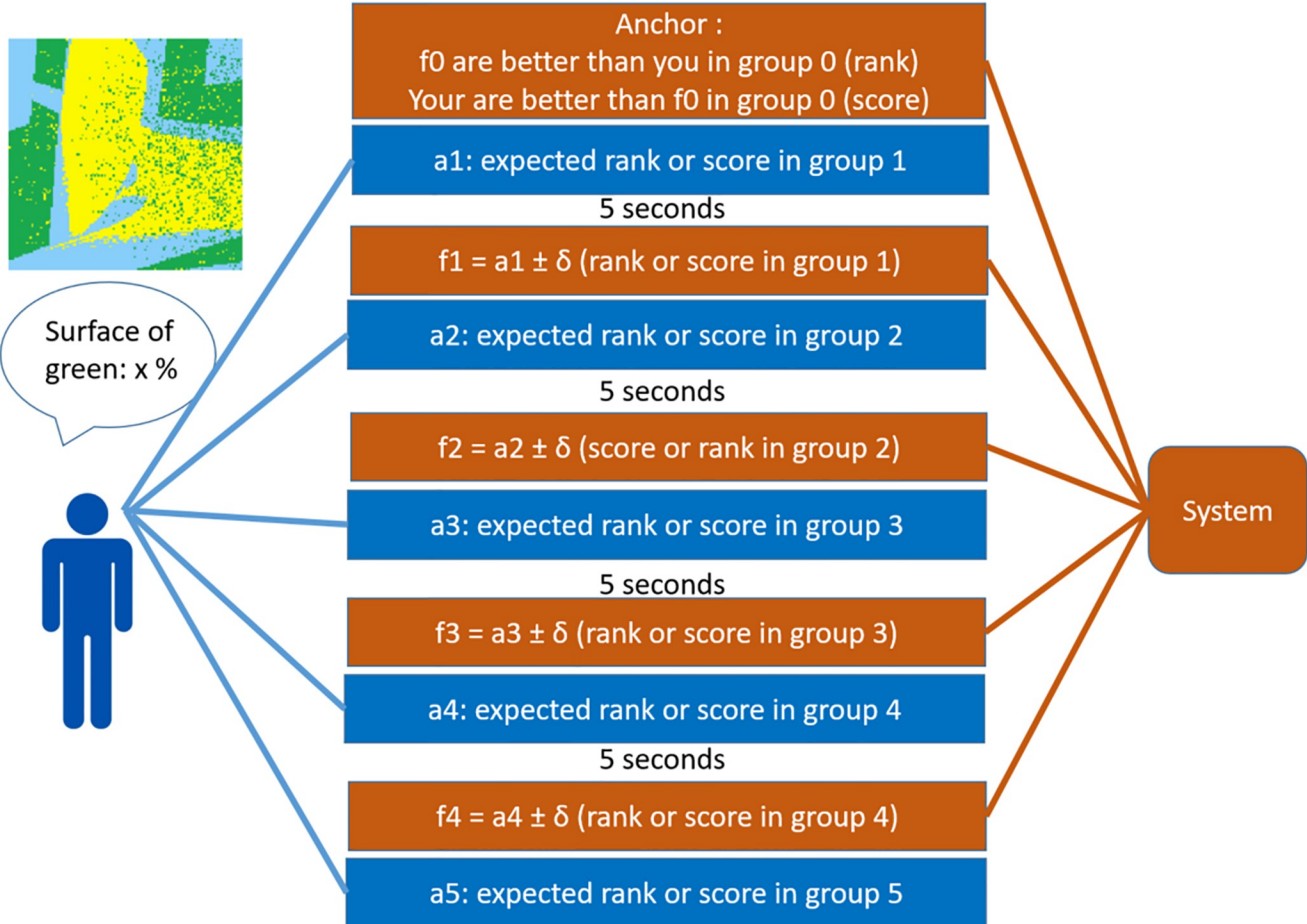

**Fig 1. Schema of the experiment.** The participant assesses the part of surface in green in images like the one on the top left of the figure. Then, the participants express their expectation of rank or score with respect to random sets of people who (allegedly) already performed the task. The feedbacks are allegedly these ranks or scores. They are actually automatically defined by the system.

the errors of the first group of randomly chosen 100 persons. We call $f_0$ the anchor because it is the initial reference evaluation for the participant. This anchor is actually defined by the experimenters in a way that is described further.

- Given this anchor as first evaluation, we assume that the participants have a precise information about their performance at the task, enabling them to self-evaluate. With this purpose, the participants are asked to express their expected evaluation in the second group of 100 people's errors ($G_1$). We interpret this expected evaluation $a_1$ as the first self-evaluation of the participant.

- The feedback $f_1$ is presented as the evaluation of the participant in group $G_1$. It is actually defined automatically as:

$$f_1 = a_1 \pm \delta \pm \epsilon, \tag{27}$$

where $\delta = 13$ and $\epsilon = 1$. The choice of $\delta$ is constrained. It should not be too small because this would make the bias difficult to detect and not too high, because then it would make the variations difficult to believe by the participants. The addition of the small variation $\pm\epsilon$ aims at avoiding to produce too regular series of feedbacks that could undermine the confidence of the participant in the reality of the feedback. We assume that the participants judge this feedback in comparison with their self-evaluation, like in the model.

- The participants are asked their expected evaluation $a_2$ in group $G_2$. They are requested to express this self-evaluation between their previous expectation $a_1$ and the feedback $f_1$ that they just received. Doing this, we impose that the sensitivity to the feedback is between 0 and 1 for each observation. The literature (e.g. [31, 32]) and several pilot experiments that we made (not-reported here) suggest that this assumption holds in a large majority of cases. Indeed, participants often put their self-evaluation outside the interval when they do not understand well the requests or do not pay attention. Therefore, the constraint on the self-evaluation is primarily a means to limit the noise in the results. Moreover, the participants are free to choose any value within the bounds. The possible variation of this choice when the self-evaluation varies is thus constrained only in its limits, not in its direction. The main subject of our investigation is precisely the direction of this variation, which we expect to be decreasing. This direction of variation is not constrained by the experimental setting.

- The same process is repeated again three times, with feedbacks $f_2$, $f_3$ and $f_4$ that are presented as the evaluation of the participant in groups $G_2$, $G_3$ and $G_4$, and requesting the participant's expected evaluations $a_3$, $a_4$ and $a_5$ in groups $G_3$, $G_4$ and $G_5$ (interpreted as successive self-evaluations). Actually, each time, the feedbacks are computed as:

$$f_t = a_t \pm \delta \pm \epsilon, \tag{28}$$

where $a_t$ is the expected evaluation of the participant in group $G_t$ given the last feedback $f_{t-1}$ which is (allegedly) their evaluation in group $G_{t-1}$.

- Finally, the participants are asked if they believed that the feedbacks were really the evaluation of their error with respect of the errors from real groups of 100 persons or if they believed that these feedbacks were manipulated by the experimenters. The participants are requested to rate their belief between 0 (the feedbacks are fake) to 10 (the feedbacks are real). In the following, we call this answer "trust in feedback" or sometimes simply "trust" of the participant.

The sequence of positive and negative feedbacks is chosen at random in the six possible sequences that contain two positive and two negative feedbacks (see Table 1). However, in some cases, when the self-evaluation $a_t$ is close to the limit 1 or 100, the chosen feedback would leave the [1, 100] interval. In these cases, the feedback is truncated in order to remain in [1, 100]. This might lead to some sequences where the positive and negative feedbacks are not balanced. We removed these sequences from the treated results.

Finally, the experiment also includes a questionnaire evaluating the self-esteem of the participants using Rosenberg's scale [33].

**Experimental design.** The experimental design includes the following conditions:

- Low anchor (randomly chosen in [15, 40]) vs high anchor (randomly chosen in [60, 85]).

- Six possible series of feedbacks (shown on Table 1);

- Evaluation by rank vs evaluation by score:

  - The rank is the number of persons (within the considered group of 100 persons) who perform better than the participant, plus one. 1 is the best rank, 101 is the worst:

  - The score is the number of persons in the group who perform worse than the participant. 100 is the best score, 0 is the worst. This is the scale used in [18].

In total there are 24 different conditions: 2 (anchor) x 6 (feedback sequences) x 2 (scale). All the conditions have the same probability, except the high anchor, which has a higher probability than the low anchor ($\frac{2}{3}$ vs $\frac{1}{3}$). Indeed, a pilot experiment suggested that sensitivity to feedbacks decreases only when the anchor is high. Therefore it appeared important to collect more data in these conditions.

In [18] the evaluation is made by score only. However, it is important in our experiment to check that a possible decreasing sensitivity is also detected when using an evaluation by rank.

**Choice of a very specific task.** The choice of the very specific task of assessing a surface within 2D images requires justification. Indeed, our main assumption is that people tend to be less sensitive to the feedbacks when their self-evaluation is high because then they tend to be more self-confident and less prone to be influenced by others. If this assumption holds, at a first glance, the experiment is more likely to succeed if the feedbacks and self-evaluations are about a general ability than about a very specific task. Indeed, a high self-evaluation at a very specific task seems less likely to affect general self-confidence.

However, we can also assume that a high self-evaluation at a very specific task has an influence on the self-confidence related to this task only. This specific increase of self-confidence would decrease the sensitivity to feedbacks about this task only. Here, we assume that the self-confidence depends, at least partly, on the context. This seems a reasonable assumption: most people tend to be self-confident in their field of expertise and more insecure in unknown situations.

Table 1. The six series of the 4 feedbacks $f_1$, $f_2$, $f_3$ and $f_4$ (two positive, two negative).

| $f_1$ | $f_2$ | $f_3$ | $f_4$ |
|:---:|:---:|:---:|:---:|
| + | + | - | - |
| + | - | + | - |
| + | - | - | + |
| - | + | + | - |
| - | + | - | + |
| - | - | + | + |

Moreover, this assumption shows strong practical advantages. Firstly, testing it avoids serious ethical and practical difficulties in manipulating the self-evaluation about general abilities. Secondly, if the task is very specific and unknown to the participants, they initially have no idea about their evaluation at this task and they can easily believe any feedback given during the experiment.

For these reasons, we finally considered that designing the experiment about a very specific task is preferable. This task is assessing the size of a coloured surface in 3 different 2D images. An example of image is shown on Fig 1 which schematises the experiment.

### Result treatment

We report how we determined our sample size, all data exclusions, all manipulations, and all measures in the study. The R code of all treatments and the data are available at https://github. com/guillaumeDeffuant/sensitivityBias.

The experiment yields a set of triples including self-evaluation at $t$, feedback at $t$, self-evaluation at $t + 1$, denoted by $(a_t^i, f_t^i, a_{t+1}^i)$, the exponent $i$ designating the participant and $t \in \{1, 2, 3, 4\}$ the index of the successive feedbacks and self-evaluations for this participant($t$ is called time step in the following). In the experiment, $f_t^i - a_t^i = \pm\delta \pm \epsilon$. As $\epsilon$ is small, in the following text, to simplify the notation, we define $\delta_t^i = \delta \pm \epsilon$, hence $f_t^i - a_t^i = \pm\delta_t^i$. We removed participants whose series of self-evaluations got too close to 0 or to 100 because we could not then apply the planned feedbacks. In the S10 and S11 Tables also show results when removing the participants who filled the questionnaire in less than 3 minutes. Indeed, 3 minutes seems a very minimal time for carefully answering the questions. However, this filter on the participants does not change the main results.

Importantly, in order to simplify the presentation of the results, we only use the increasing scale of evaluation (score). Hence the first treatment is to transform any rank $r$ whether allegedly computed by performance comparison in a group (for feedbacks) or expected by the participant (for self-evaluations) into $100 - r$.

**Linear approximations of the sensitivity to feedback functions.** Our first aim is to check the hypothesis that sensitivities $h(a_t)$ to all feedbacks, $h_p(a_t)$ to positive feedbacks and $h_n(a_t)$ to negative feedbacks are decreasing.

The ideal approach would be to derive from the data of each participant $i$, approximations of the sensitivities $h^i(a_t)$ to all feedbacks, $h_p^i(a_t)$ to positive feedbacks and $h_n^i(a_t)$ to negative feedbacks of this participant. However, there are only four triples $(a_t^i, f_t^i, a_{t+1}^i)$ available for each participant and this is not enough to get a reliable approximation.

Instead of computing the sensitivities of a single participant, we derive approximations of the sensitivities from samples of triples $(a_t^i, f_t^i, a_{t+1}^i)$ mixing several participants and several time steps. The larger size of the sample provides higher chances to get significant results. We assume that, in this case, we obtain an approximation of the average sensitivities to feedbacks of the participants in the set.

Hence, considering a sample $A$ of triples $(a_t^i, f_t^i, a_{t+1}^i)$ mixing participants and time steps, we compute the linear regressions taking self-evaluation change $\left|\frac{a_{t+1}^i - a_t^i}{\delta_t^i}\right|$ as outcome variable and self-evaluation $a_t^i$ as predictor variable and this provides linear approximations of the sensitivities $h$, $h_p$ and $h_n$ defined in Eqs 1, 19 and 20. More precisely, from a given sample of triples $(a_t^i, f_t^i, a_{t+1}^i)$ we derive the following linear models:

- For all feedbacks $f_t^i = a_t^i \pm \delta_t^i$, the linear model:

$$\frac{|a_{t+1}^i - a_t^i|}{\delta_t^i} \approx c\frac{a_t^i}{100} + b \approx h(a_t^i), \tag{29}$$

approximates the sensitivity to feedbacks $h$;

- For positive feedbacks $f_t^i = a_t^i + \delta_t^i$, the linear model:

$$\frac{|a_{t+1}^i - a_t^i|}{\delta_t^i} \approx c_p\frac{a_t^i}{100} + b_p \approx h_p(a_t^i), \tag{30}$$

approximates the sensitivity to positive feedbacks $h_p$;

- For negative feedbacks $f_t^i = a_t^i - \delta_t^i$, the linear model:

$$\frac{|a_{t+1}^i - a_t^i|}{\delta_t^i} \approx c_n\frac{a_t^i}{100} + b_n \approx h_n(a_t^i), \tag{31}$$

approximates the sensitivity to negative feedbacks $h_n$.

In the following, these linear models are respectively called the sensitivity to feedbacks, the sensitivity to positive feedbacks and the sensitivity to negative feedbacks in sample $A$. The sign of slopes $c$, $c_p$ and $c_n$ indicates if these functions are increasing or decreasing.

These linear regressions are computed in various subsets of the whole set of triples, mixing more or less participants and time steps.

When the sets include 3 or 4 time steps, we also computed linear mixed effect models, using the R package lme4 [34] for the approximation of the sensitivity to feedbacks (positive and negative together), because the 3 or 4 self-evaluations of a single participant are not independent [35]. In the other cases, sets including less than 3 time steps, or including only positive or only negative feedbacks, the linear mixed effect model is not applicable and we use only standard linear regressions (see more details in the S1 Appendix).

**Total bias $B$ in a sample.**   The measure of the total bias $B(A)$ is performed on a sample $A$ such that the triples $(a_t^i, f_t^i, a_{t+1}^i)$, for the four time steps $t \in \{1, 2, 3, 4\}$ are included in $A$, for each of participant $i$. Formally, the total bias $B(A)$ is:

$$B(A) = \frac{1}{p_A}\sum_i \frac{a_5^i - a_1^i}{\frac{1}{2}\sum_{t \in \{1,\dots,4\}}\delta_t^i}, \tag{32}$$

where $p_A$ is the number of participants in sample $A$.

Indeed, $B(A)$ is the average difference between the last self-evaluation ($a_5$) of the series of two positive and two negative feedbacks and the first one ($a_1$), as a proportion of the average feedback intensity in the series of four triples. Hence, we divide $a_5^i - a_1^i$ by twice the average of feedback intensity $\delta_t^i$ for $t = 1, \%, 4$. Moreover, $B(A)$ is the sum of the self-enhancement bias and the bias from decreasing sensitivity as shown by Eq 24.

**Average self-enhancement bias $E$ in a sample.**   The evaluation of the self-enhancement bias in a sample of triples $A$ is based on the sensitivity to positive and to negative feedbacks in this set. It is the average of the self-enhancement for $a_t^i$ as defined by Eq 18, i.e. the difference between the reaction to a positive and to a negative feedback at $a_t^i$. Let $(c_p, b_p)$ and $(c_n, b_n)$ be the slope and intercept of respectively the approximate sensitivity to positive and negative feedbacks, as defined by Eqs 30 and 31. Applying Eq 18 with these approximate functions and

averaging on sample $A$ yields:

$$E(A) = \frac{1}{n_A} \sum_{i,t} (c_p - c_n) \frac{a_t^i}{100} + b_p - b_n, \tag{33}$$

where $n_A$ is the number of triples in sample $A$.

The self-enhancement bias $E$ can be seen as the average change of self-evaluation after a sequence of two opposite feedbacks, without taking into account the variation of sensitivity to the feedback. This change is a proportion of the feedback intensity (here since $\delta_t^i = |f_t - a_t| = \delta \pm \epsilon$, it is roughly a proportion of $\delta$).

The self-enhancement bias is negative when the participants are, on average, more sensitive to the negative than to the positive feedbacks.

**Theoretical bias from sensitivity of feedbacks $S'$.**   This measure is the theoretical average change of self-evaluation due to the decreasing sensitivity to feedbacks. Following formula 26, this measure is:

$$S'(A) = \frac{1}{n_A} \sum_{i,t} - c_m \left( c_m \frac{a_t^i}{100} + b_m \right) \delta_t^i, \tag{34}$$

where $c_m = \frac{c_p + c_n}{2}$ and $b_m = \frac{b_p + b_n}{2}$. Note that we divided Eq 26 by $\delta_t^i$, so that this measure is expressed as a percentage of $\delta_t^i$ like self-enhancement and total biases. The difference between total and self-enhancement biases is the bias from sensitivity and it should be close to the theoretical value:

$$S(A) = B(A) - E(A) \approx S'(A). \tag{35}$$

Fig 2 illustrates the computation of the biases from self-enhancement and sensitivity. Note that both the self-enhancement bias and the theoretical bias from sensitivity can be computed

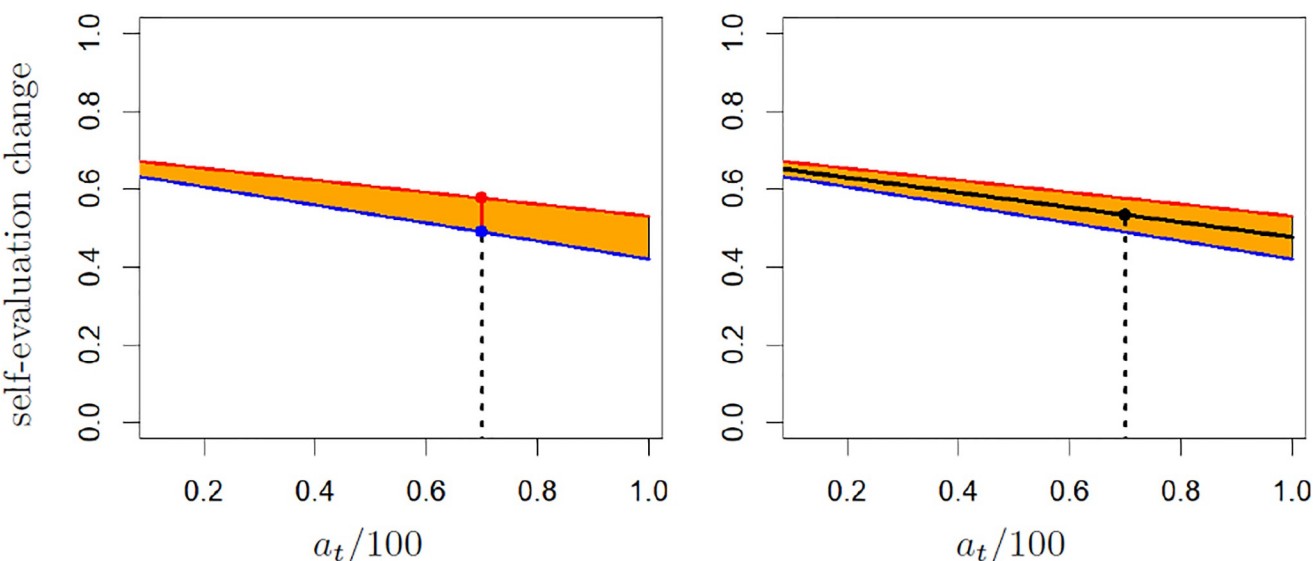

**Fig 2. Illustration of the computation of biases from self-enhancement and from sensitivity.** The self-evaluation $a_t/100 = 0.7$ is shown by the dotted line. In both panels, the red and blue lines are the linear approximation of the sensitivity to respectively the positive and negative feedbacks. On the left panel, the self-enhancement for $a_t$ is the ordinate of the projection of point $(a_t/100, 0)$ on the red line (red point) minus the ordinate of the projection of point $(a_t/100, 0)$ on the blue line (blue point). On the right panel, the bias from sensitivity is $-c_m(c_m(a_t/100) + b_m)\delta$, where $c_m x + b_m = 0$ is the equation of the medium line between the red and blue lines.

as soon as the sensitivity to positive and to negative feedbacks are available. In particular, they can be computed on sets including incomplete series of triples for each participants (i.e. including less than the four time steps).

**Bootstrap.** Bootstrap is used to evaluate the variability of a quantity $Q$ that is derived from a sample $A$ [36]. Its principle is to generate a large number of random samples (with replacement) $A^k$ of $A$, each of them providing an evaluation of $Q^k$ of the quantity. Statistics on the values $Q^k$, for instance quantiles or standard deviation, provide an evaluation of the variability of $Q$ because of the sample. We use bootstrap in order to evaluate the variability of the different measures (enhancement bias, sensitivity bias, theoretical sensitivity bias).

Moreover, when the considered sample $A$ includes the triples $(a_t^i, f_t^i, a_{t+1}^i)$, for the four time steps $t \in \{1, 2, 3, 4\}$, for each of participant $i$ in set $A$, then bootstrap helps to evaluate the robustness of the difference between the sensitivity bias (measured as the difference between the total bias and the enhancement bias) and the theoretical sensitivity bias (measured as the average of the sensitivity bias on the sample). If we constitute the samples $A^k$ as usual by randomly choosing triples $(a_t^i, f_t^i, a_{t+1}^i)$, in general, we do not keep the sequences of the four time steps $t \in \{1, 2, 3, 4\}$ complete, thus there is no guarantee to be able to compute the total bias $a_5^i - a_1^i$. Therefore, in this case, instead of deriving the sample $A^k$ by drawing triples $(a_t^i, f_t^i, a_{t+1}^i)$ in $A$ at random, we draw the participants at random (with replacement) and for each drawn participant $i$, we add the triples $(a_t^i, f_t^i, a_{t+1}^i)$, for the four time steps $t \in \{1, 2, 3, 4\}$ into the new sample $A^k$.

In the following we compute means and standard deviation on 200 bootstrap samples. We checked on several sets that, with this number of samples, the mean and standard deviation are at most around 20% different from the values obtained with 1000 samples. We considered that this level of precision is sufficient.

The bootstrap allows us to compute effect size when comparing the results of a measure on two sets $A_1$ and $A_2$. The effect size is indeed defined as:

$$s = \frac{|m_1 - m_2|}{\sigma_1}, \tag{36}$$

where $m_1$ and $m_2$ are the bootstrap averages of the considered measure computed respectively on $A_1$ and $A_2$, and $\sigma_1$ is the bootstrap standard deviation of the measure for set $A_1$. An effect size of 0.2 is small, of 0.5 medium, of 0.8 large and of 1.3 very large [37].

**Power analysis.** The power analysis concerns the linear regressions that approximate the sensitivity to feedbacks in different data sets. Using the G*Power software for linear regressions, with one tail, seeking a small effect (0.1 recommended by the software) and a power ($1 - \alpha$) of 0.95, we get a recommended sample size of 1073.

We need to perform linear regressions for both positive and negative feedbacks and at least when considering each scale independently. This leads to a recommended sample size of 4*1073 = 4302.

We decided to collect a sample from 1500 participants, generating, with 4 triples each, a sample of size 4*1500 = 6000. The surplus of around 30% accounts for the unavoidable unreliable data (participants not believing in the feedbacks or with self-evaluations that make the feedbacks leave the interval [1, 100]).

## Results

The experiment involves 1509 participants (803 females, 706 males, age between 17 and 79). We removed 141 participants because their series of self-evaluations got too close to 0 or 100 and we could not apply the planned feedback. In total, after these exclusions the data set

includes 5472 triples $(a_t^i, f_t^i, a_{t+1}^i)$ for 1368 participants (729 women and 639 men, mean age: 36.8 years). The data of the experiment and all the results are available at https://github.com/guillaumeDeffuant/sensitivityBias.

## Checking main hypotheses

**The sensitivity to feedback decreases when self-evaluation increases.**   Our first hypothesis is that the sensitivity to feedbacks decreases when self-evaluation increases. This sensitivity is evaluated by the linear regression taking change of self-evaluation as outcome variable and self-evaluation as predictor variable (Eq 29), that we recall here for convenience:

$$\frac{|a_{t+1}^i - a_t^i|}{\delta_t^i} \approx c\frac{a_t^i}{100} + b. \tag{37}$$

We also evaluate the slope $c$ using a linear mixed effect model [34, 35], which takes the non-independence of the self-evaluations from the same participants into account (see S1 Appendix for details).

Table 2 shows the value of the slope of the sensitivity $c$ derived from different data sets mixing several time steps $((1:k) = \{1, .., k\})$ and participants of different values of trust in the feedback (on the first line of the table all the possible values of trust are considered). For the three first time steps or all the four time steps ($t \in (1:3)$ or $t \in (1:4)$), the table shows the slope computed with the linear mixed effect model ($c$ (lmer)). Note that the method does not provide a p-value associated with the slope. The table also shows the size $N$ of each data set.

In all cases, the value of the slope $c$ is significantly negative, with an increasing amplitude when trust in the feedback increases. The slopes computed with the linear mixed effect model are very close to the slopes obtained by standard linear regression (only slightly less negative). Therefore, the non-independence of the self-evaluations for each participant has a weak effect.

Overall these results confirm our main hypothesis that sensitivity to feedback decreases when self-evaluation increases. The confirmation is particularly clear in sets of participants reporting a high trust, but the hypothesis is also valid in those reporting a low trust. Fig 3 represents two examples of sensitivity to feedbacks computed on sets of participants with high trust.

**Table 2. Slope of sensitivity to feedback on several time steps.** The slope is computed for different intervals of trust. $N$ is the number of triples $(a_t^i, f_t^i, a_{t+1}^i)$ in the considered data set. $c$ is the slope computed with a standard linear regression and $c$(lmer) designates the slope computed with a linear mixed effect model (p-value not provided).

| Slope of sensitivity to feedback on several time steps | | | | | | | | |
|---|---|---|---|---|---|---|---|---|
| **Trust** | **$t \in (1:2)$** | | **$t \in (1:3)$** | | | **$t \in (1:4)$** | | |
| | $N$ | $c$ | $N$ | $c$ | $c$(lmer) | $N$ | $c$ | $c$(lmer) |
| [0, 10] | 2736 | −0.08** | 4104 | −0.09*** | −0.08 | 5472 | −0.07*** | −0.06 |
| [0, 6] | 1656 | −0.07* | 2484 | −0.07* | −0.07 | 3312 | −0.04 . | −0.04 |
| [7, 10] | 1080 | −0.12** | 1620 | −0.13*** | −0.12 | 2160 | −0.11*** | −0.1 |
| [8, 10] | 834 | −0.16*** | 1251 | −0.15*** | −0.14 | 1668 | −0.13*** | −0.12 |
| [9, 10] | 562 | −0.18** | 843 | −0.18*** | −0.17 | 1124 | −0.16*** | −0.14 |

\*\*\* : $p < 0.001$,

\*\* : $p < 0.01$,

\* : $p < 0.05$,

. : $p < 0.1$

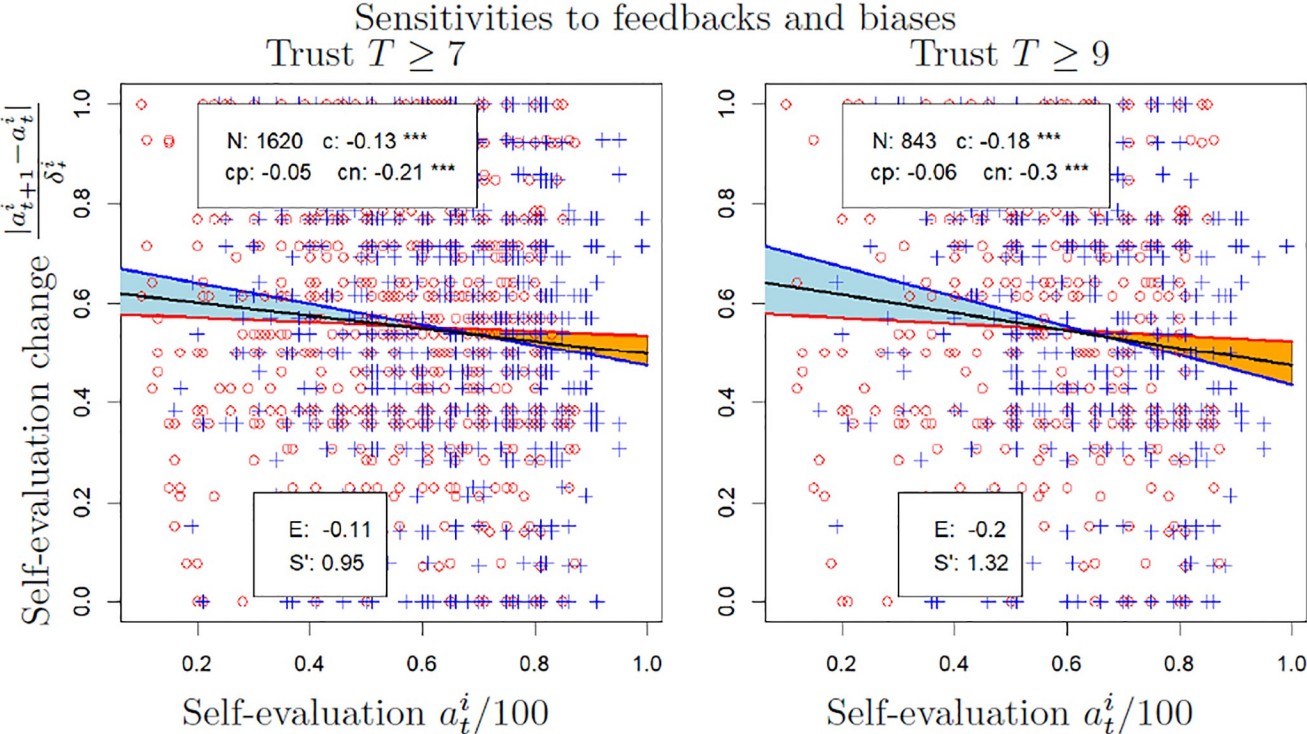

**Fig 3. Examples of regressions and measures of the different biases.** The values of the self-evaluation change axis are $\frac{|a_{t+1}^i - a_t^i|}{\delta_t^i}$ and those of the self-evaluation axis are $a_t^i/100$. The black line is the regression computed on the whole set (of slope $c$), the red line for positive feedbacks (of slope $c_p$) and the blue line for negative feedbacks (of slope $c_n$). The red points represent the couples (self-evaluation, self-evaluation change) for positive feedbacks, the blue points represent those couples for negative feedbacks. The orange surface represents positive self-enhancement while the light blue surface represents negative self-enhancement (self-derogation). $E$ is the average self-enhancement bias and $S'$ is the estimation of the average bias from sensitivity to feedbacks, both expressed as a percentage of $\delta$.

The values of $c$ are slightly less negative for $t \in (1:4)$, especially when participants report a low trust. This suggests to compute the sensitivity to feedbacks on data from each time step as shown by Table 3. This table suggests that the behaviour of the participants is stable on average on the three first time steps and changes significantly at $t = 4$. Indeed, the slope of the sensitivity $c$ is significantly negative ($p$-value at least $<0.1$) except at the last time step ($t = 4$) and for participants reporting low trust ($T \le 6$) and $t > 1$. The slope $c$ is less significant than in Table 2 which could be expected as the data sets are smaller.

A possible explanation for the change of behaviour at $t = 4$ is a loss of attention after repeating the same process of evaluations many times.

The interested reader can find complementary results about slopes $c_p$ of the sensitivity to positive and $c_n$ of the sensitivity to negative feedbacks in the (S1 Table).

**The decreasing sensitivity to feedback generates a measurable positive bias.** Our second hypothesis is that the decreasing sensitivity to feedback generates a measurable positive bias in self-evaluation. As presented in the section devoted to the result treatments, on a data set $A$ mixing the four time steps ($t \in (1:4)$), we have two ways of measuring this bias:

- The difference between the total bias ($a_5 - a_1$) and the self-enhancement bias, denoted by $S$ ($A$) (Eq 35). This measurement is relevant only for data sets that include the four time steps;

**Table 3. Slope $c(t)$ of sensitivity to feedback at each time step.** The slope is computed on data from participants reporting different trust values (first column). $N$ is the number of triples $(a_t^i, f_t^i, a_{t+1}^i)$ in the considered data sets (it does not vary with $t$).

| | | Slope $c(t)$ of sensitivity to feedback at each time step | | | |
|---|---|---|---|---|---|
| Trust | $N$ | $c(1)$ | $c(2)$ | $c(3)$ | $c(4)$ |
| [0, 10] | 1368 | −0.09* | −0.07 . | −0.09* | −0.01 |
| [0, 6] | 828 | −0.08 . | −0.06 | −0.06 | 0.02 |
| [7, 10] | 540 | −0.13* | −0.12* | −0.13* | −0.07 |
| [8, 10] | 417 | −0.14* | −0.18** | −0.13 . | −0.1 |
| [9, 10] | 281 | −0.2* | −0.16 . | −0.19* | −0.09 |

** : $p < 0.01$,

* : $p < 0.05$,

. : $p < 0.1$

- The average of the sensitivity bias measured from the sensitivities to positive and negative feedbacks, denoted by $S'(A)$ (Eq 34). The sensitivities to positive or negative feedbacks can be computed only with standard linear regressions because the sets include at most two time steps for which the feedback is of the same sign, which makes the linear mixed effect model non-applicable.

Fig 4 shows the average and standard deviation of these measures, computed on 200 bootstrap samples, for $t \in (1 : 4)$ and for the different values of trust considered in the previous tables. The values of these biases are percentages of the feedback intensity $\delta$, hence a value around 1% should therefore be interpreted as an average increase of the self-evaluation of 1% of $\delta$ at each time step.

These results suggest that a bias from sensitivity is detected with both measurements in all cases except for participants reporting low trust (interval [0, 6]). Indeed, except for this set of low trust, the standard deviation on the bootstrap is around one third of the mean. Therefore, the bias is not likely to be 0. In the set of participants reporting trust in the interval [0, 6], the standard deviation is close to the mean, therefore the value is not significant. Note that both $S$ and $S'$ increase when trust increases.

Moreover, the small difference between $S$ and $S'$ indicates that $S'$ is a reliable approximation of $S$ on sets where $S$ cannot be computed (because the data include values from only a part of the four time steps). This allows us to measure the bias from sensitivity to feedbacks on data for time steps in (1 : 2) and (1 : 3). This is important because we noticed previously that the data for $t = 4$ are probably of lower quality, thus the measures on sets excluding the last time step are likely to be more accurate.

Fig 5 shows the values of the measure $S'$ of the bias from sensitivity for time steps (1 : 2) and (1 : 3). We observe that in this case, even for participants reporting low trust, a significant bias is detected as the standard deviation is lower than half the mean. Moreover, as expected from the values of the slope $c$ of the sensitivity to the feedbacks, the values of the bias are higher than for $t \in (1 : 4)$. This can be explained again by the change of behaviour at $t = 4$.

Fig 3 illustrates the results obtained on sets of participants reporting a level of trust higher than 7 or higher than 9 and for $t \in (1 : 3)$. In both cases, the self-enhancement is negative (light-blue surface) when the self-evaluation is low and positive (orange surface) when the self-evaluation is high.

If the data at $t = 4$ are considered as unreliable, Fig 5 provides the relevant measurements of the bias from sensitivity to feedbacks. In this case, the results show again that a bias from

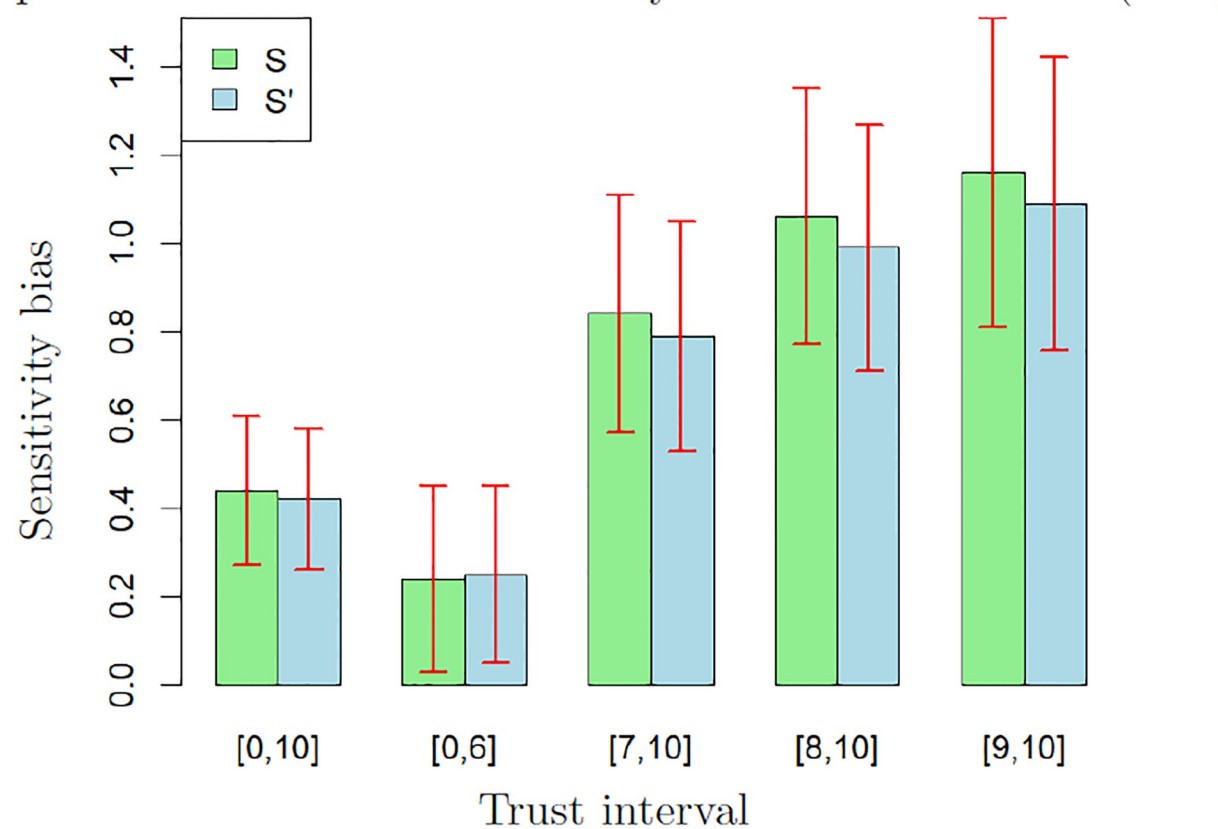

**Fig 4. Comparison of measures $S$ and $S'$ of the bias from sensitivity to feedbacks for $t \in (1:4)$.** Both measures are computed on data from participants reporting trust in an interval. The value and the error bars are the average and standard deviation computed on 200 bootstrap samples.

sensitivity is detected for all the tested levels of trust (even in the trust interval $[0, 6]$), and its value increases with trust. This confirms our second hypothesis.

### Variations of biases with scale, gender and self-esteem

We focus on the participants reporting a high level of trust ($T \in [7, 10]$), as the data from these participants are the most meaningful. The results for low trust ($T \in [0, 6]$) are available in the Supplementary materials. Figs 6 and 7 show the mean and standard deviation of sensitivity and self-enhancement biases computed over 200 bootstrap samples from sets differentiating scales (rank or score) gender and self-esteem (SE in the graph). The values are shown for $t \in (1 : 3)$ which we consider as the most relevant cases. The results for $t \in (1 : 2)$ and for $t \in (1 : 4)$ show broadly the same features (they are available in Supplementary Materials). Moreover, the variations of the sensitivity bias with the anchor, which seem to us more peripheral, are available in the S2 and S3 Tables.

Fig 6 shows that the bias from sensitivity varies around 1% of the feedback intensity. Except for the set of participants of low self-esteem which shows a sensitivity bias close to 0, the bias for other sets varies between 0.73% (male with evaluation by rank) to 1.34% of the feedback (male with evaluation by score). For participants of high self esteem and women, the difference between evaluation by rank and by score is weak (effect size 0.12 and 0.07 respectively). The

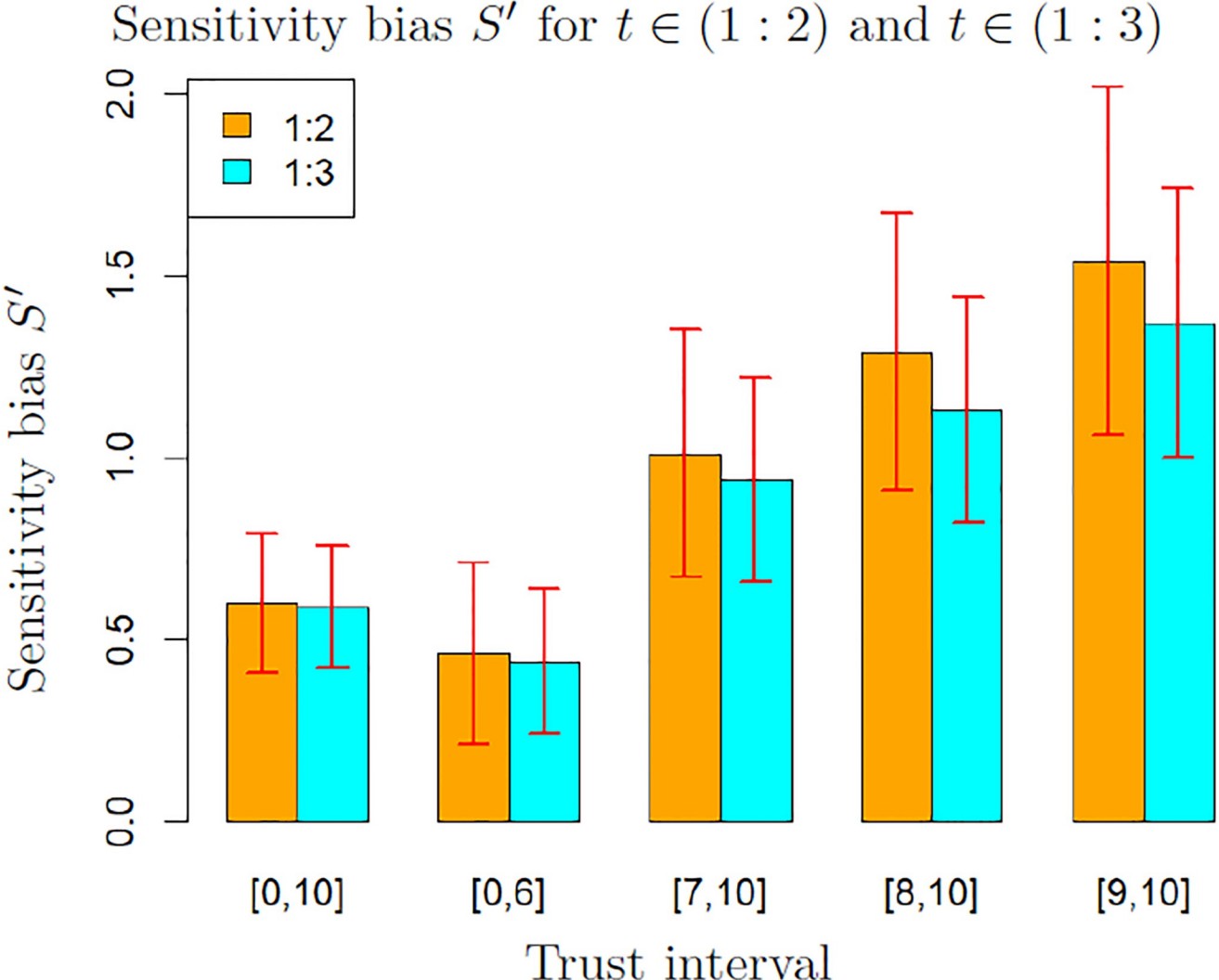

**Fig 5. Measure $S'$ of the bias from sensitivity to feedbacks for $t \in (1:2)$ and $t \in (1:3)$.** The values and the error bars are the mean and standard deviation computed on 200 bootstrap samples.

relative stability of the sensitivity bias is remarkable as the closely associated self-enhancement bias shows much stronger variations, as shown on Fig 7:

- The self-enhancement bias changes dramatically with the scale: it is significantly positive when participants self-evaluate by rank and significantly negative when they self-evaluate by score in all considered sets (effect size 7.86 between rank and score).

- The self-enhancement bias is higher for men and for participants with high self-esteem than for women and for participants with low self-esteem. The difference is very significant when participants self-evaluate by score (effect size 2.84 between men and women and 3.91 between participants with high self-esteem and women). In the whole data set, the average self-esteem of men (3.08) is only slightly higher than the average self-esteem of women (2.98) and this difference of self-esteem seems insufficient to explain the strong difference of self-enhancement bias between men and women.

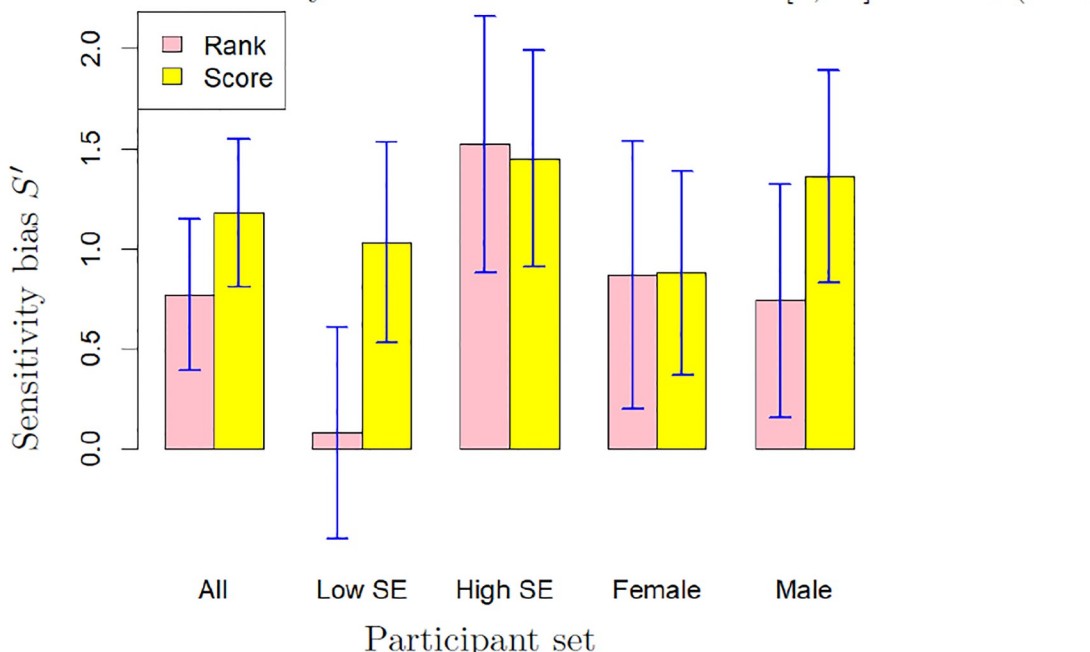

**Fig 6. Sensitivity bias *E* for different values of scale, gender and self-esteem (low SE: Self-esteem ≤3, high SE: Self-esteem >3), for trust in [7, 10] and** $t \in (1:3)$**.** The bars and the error bars are respectively the mean and standard deviation computed on 200 bootstrap samples.

## Discussion

We first discuss the results on the variations of self-enhancement and self-derogation and then the results about the decreasing sensitivity to feedbacks and its associated bias.

### Comments on the variations of self-enhancement bias

This paper focusing on the bias from decreasing sensitivity, we limit ourselves to preliminary comments and remarks about self-enhancement bias.

**Effect of scale.** The very significant self-derogation observed when participants self-evaluate by score deserves a specific discussion. This result is in line with the significant negative bias observed in the experiment reported in [18], which shows strong similarities with ours. However, in [18] this negative bias is explained by the motive of participants to improve their results at the task, which incites them to draw more attention to more informative negative feedbacks. This explanation seems irrelevant in the context of our experiment because the participants cannot improve their performance at the task, which is achieved at the beginning of the questionnaire and cannot change afterwards. Arguing that the participants are learning to self-evaluate could be a way to introduce the learning context. However, we should then observe self-derogation when the evaluation is made by rank as well, but it is not the case at all, as we measure a significantly positive self-enhancement bias in all sets of participants self-evaluating by rank. These results suggest an effect of the scale.

We can only formulate a preliminary hypothesis that an evaluation using a growing scale could be perceived as a possessed quantity, as much as the evaluation of an ability. Then, a

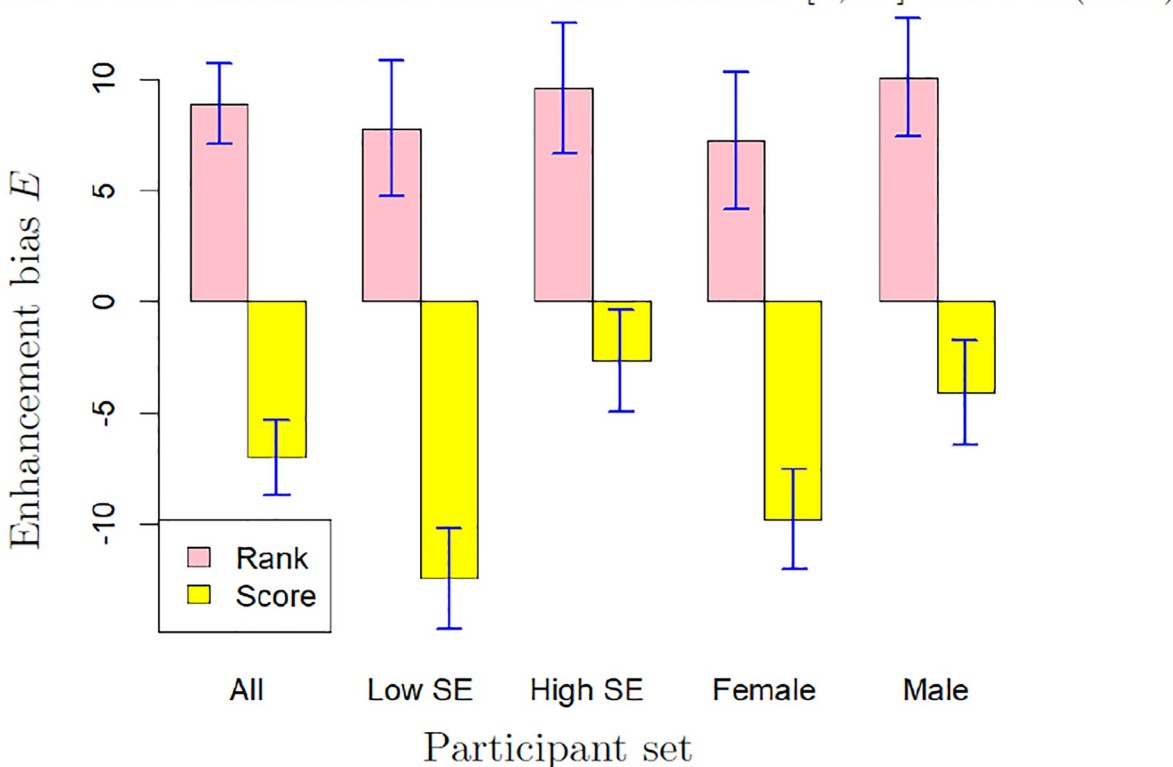

**Fig 7. Enhancement bias *E* for different values of scale, gender and self-esteem (low SE: Self-esteem ≤3, high SE: Self-esteem >3), for trust in [7, 10]** and $t \in (1:3)$**.** The bars and the error bars are respectively the mean and standard deviation computed on 200 bootstrap samples.

negative feedback would be perceived as the loss of some possessions as well as a set back in status. The higher reaction to negative feedbacks could then be related to a general loss aversion [38] or higher sensitivity to negative events [39]. By contrast, the evaluation by rank seems to be more exclusively related to a perceived status, triggering self-enhancement, as expected from the literature.

A possibly related effect of the scale could be found in the experiment reported in [40], suggesting that student performance is significantly improved when using a grading system based on student ranking rather than on performance standards. Our results suggest that the grading systems generate significantly different self-enhancement or self-derogation biases, which could influence the performance of the students. In particular, strong levels of self-derogation which, extrapolating from our results, could be expected with the grades based on performance standards, could discourage students. Of course, this does not exclude the influence of other factors mentioned in [40].

**Effect of self-esteem.** Self-enhancement and self-esteem are deeply related as the self-enhancement motive is to preserve or increase self-esteem. Yet, the literature shows contradictory views about the influence of self-esteem on self-enhancement [41]. The self-enhancement theory for instance assumes that individuals with a low self-esteem have a stronger motivation for self-enhancement [42]. Other theories suggest that, on the contrary, individuals with a high self-esteem are more motivated to protect their positive self-view [22, 23] or to confirm it [18]. The experiments reported in [24] corroborate the latter theories, as they suggest that the status

of expert may provide enough overconfidence to claim impossible knowledge. The experiment reported in [18] corroborates them as well, as participants with a lower self-esteem tend to show a more significant negative bias. Similarly, in our results, sets of participants with a lower self-esteem tend to show a greater self-derogation (for the evaluation by score) and a lower self-enhancement (for the evaluation by rank).

Moreover, this tendency is also confirmed when considering how the self-enhancement varies when self-evaluation increases within a set, instead of comparing the average measures from different sets. Indeed, within most considered sets, self-enhancement increases when self-evaluation increases, because the slope of the sensitivity to negative feedbacks is more negative than the slope of the sensitivity to positive feedbacks in most of the sets (S1 Table).

**Effect of gender.** Previous research established gender differences in self-enhancement. First, men tend to engage more, comparatively to women, in self-deceptive enhancement and women more in impression management [29]. Our experiment does not involve much impression management, as the participants are told that they interact, via the computer, with a program that computes their rank or score in several predefined groups. This can explain the lower self-enhancement of women measured in our experiment.

Also, the gender difference may depend on the context. In general, men reveal significantly higher self-enhancement with respect to masculine subjects than women do, whereas the self-enhancement of men and women in relation to feminine subjects are similar [25, 26]. More generally, men tend to show a higher self-enhancement than women in a context where qualities related to agency (competence, independence, openness) are important [27] as opposed to qualities related to communion (warmth, interdependence, agreeableness) [28].

Arguably, the task of surface assessment in our experiment can be perceived as close to mathematics, a masculine subject, and the self-evaluation concerns an individual competence. This can explain that men show a higher self-enhancement in our results when they self-evaluate by rank. This explanation seems more dubious for the lower self-derogation of men when participants self-evaluate by score. Indeed, if our hypothesis that the score is also perceived as a possessed quantity which decreases in case of negative feedbacks, the gender difference in self-derogation should probably be rather related to gender differences with respect to loss aversion or to perception of negative events.

## Discussion about the bias from decreasing sensitivity to feedback

The results support our first main hypothesis that the sensitivity to feedbacks decreases with the self-evaluation. Indeed, we measure a significant decrease of sensitivity to feedbacks in the set of all the participants and in sets of participants of different trust in the feedbacks. The decrease is more significant in sets of participants reporting high trust and when excluding the last time step.

The results also support our second main hypothesis because we detect a significant positive bias from this decrease of sensitivity, which is added to the usual self-enhancement bias. As expected, this bias is more significant in sets of participants reporting high trust, because the sensitivity to feedbacks decreases more significantly in these sets. The bias is around 1% of the feedback intensity in these sets and appears rather stable when scale, self-esteem or gender vary. We now discuss the general significance of the newly detected bias in relation to the motivations for self-enhancement and self-assessment. In [16], self-enhancement and self-assessment are defined as follows:

- "self-enhancement is the motivation of people to elevate the positivity of their self-conceptions and to protect their self-concepts from negative information,

- self-assessment is the motivation of people to obtain a consensually accurate evaluation of the self."

Moreover, [16] stresses that the positive bias on self-evaluation induced by self-enhancement is often considered useful because it can provide the will or general self-efficacy necessary to initiate novel action. As expressed by [43]: "Even if one is sick and anxious and poor, there should be reason to get up in the morning. . .Hence self-cognitions do not always have to be veridical in order to be functional".

However, excessive self-overestimation can expose to severe negative consequences as shown in various domains such as health, education and the workplace [1]. Moreover, it can lead to excessive narcissism [28] or bitterness when people become the only ones convinced of their own high merit.

The motivation for self-assessment can be seen as contradicting self-enhancement. Indeed, self-assessment removes the protection against negative feedbacks in order to get an unbiased and accurate self-perception. There is thus a tension between both motivations as, in principle, an accurate self-assessment should remove the positive bias from self-enhancement.

This work suggests that, when the sensitivity to feedbacks decreases as the self-evaluation increases, the self-assessment process, though removing protections against negative feedbacks, also generates a positive bias. Consider situations where the feedbacks fluctuate around an fixed average value, at least for a while. These situations seem indeed more likely in everyday life than series of feedbacks of alternating intensities. As shown at the beginning of the "Material and methods" section, in these situations, the decreasing sensitivity to feedbacks implies an average self-evaluation that is slightly higher than the average feedback. Then, in some cases, this higher self-evaluation influences the average feedback after a while. If the average feedback increases, then the average self-evaluation increases again, and so on. However, if the average feedback significantly decreases, then the self-evaluation adapts and decreases as well (though remaining slightly higher than the new average). Therefore, the bias from decreasing sensitivity pushes the subject forward, in a cautious and adaptive way.

Moreover, the bias from decreasing sensitivity is more likely to take place when people accumulate many feedbacks. Indeed, as the bias is an average, it is likely to be wrongly appreciated on a low number of feedbacks. Therefore, active and daring people who are eager to accumulate experiences benefit from it more regularly. In comparison, a significant bias from self-enhancement can appear by dismissing a small number of negative feedbacks and the self-overestimation may then remain more or less stable, even if the average feedback decreases.

Let us illustrate these remarks with an example. Tennis players of a given level tend to loose against players of superior level and win against ones of inferior level. Assuming that their sensitivity to wins and losses and to successes and failures of their shots decreases with their self-evaluation, the players are subject to the positive bias from decreasing sensitivity. If their motivation of self-assessment is very high and they self-evaluate without self-enhancement (i.e. giving the same weight to successes and failures in their shots and their matches) they tend to self-evaluate a bit higher than their actual level. This slight surplus of confidence is likely to help them win tight matches against opponents of similar level because they are likely to remain positive in difficult situations and take reasonable risks. These wins actually increase slightly their level. Hence, their self-evaluation is likely to increase as well, and so on. However, if their self-evaluation raises too much, the players are likely to experience more frequent losses against players that they consider inferior and they will decrease their self-evaluation accordingly (still keeping it a bit higher than their average level).

By contrast, players who self-evaluate with self-enhancement tend to overestimate their level even if they do not play much, because they blame external conditions (the racket, the

balls, the wind, the public. . .) for their failures. Then, they often poorly adjust their game during matches, because they overestimate their own shots and underestimate the ones of their opponents. Moreover, their inability to adjust their self-evaluation from even more losses increases the discrepancy.

This example considers the theoretical case of a player succeeding to remove any self-enhancement bias. However, our experiment suggests that in most cases, the bias from sensitivity only marginally modifies a significantly bigger self-enhancement bias. The conclusion is therefore normative: self-assessing as honestly as possible, removing any self-enhancement or self-derogation, is a recommended goal because the bias from sensitivity provides a small over-estimation that helps being positive and active without taking disproportionate risks. This conclusion complements previous research about "optimal self-esteem" [44], which distinguishes between fragile and secure self-esteem and advocates for unbiased processing in order to reach authenticity. Indeed, our work suggests that the effort for unbiased processing makes more relevant the slight supplement of positive self-evaluation generated by the decrease of sensitivity to feedbacks.

## Limitations and future challenges

A major limitation of the experiment is the small size of data obtained from each participant. Indeed, the sensitivity to feedback may vary significantly with the participants and computing average sensitivity functions in sets of participants hides this variety. Ideally, the experiment should collect a much larger number of triples (self-evaluation, feedback, change in self-evaluation) from each participant in order to derive significant individual models of sensitivity to feedbacks. Moreover, in the experiment, we constrained the self-evaluations to be between the previous self-evaluation and the feedback. We underlined that this restriction does not constrain the change of sensitivity when the self-evaluation varies, therefore, as we find a decrease of this sensitivity when self-evaluation increases with the restriction, we should also find it without the restriction. Nevertheless, it would be important to check this assumption by replicating the experiment without the restriction on the self-evaluations. This would certainly require a larger sample to cope with more noisy data.

A second limitation worth mentioning relates to the detection of the sensitivity bias itself. Indeed, its detection in a very specific experiment says nothing about its potential role in daily life. Actually, our measures suggest that the sensitivity bias might generally be too dominated by self-enhancement or self-derogation biases to show an independent effect. Moreover, the agent based model simulations suggest that the sensitivity bias has a significant effect only on the long term. In this case again, long series of data could provide more information on the potential effect of this bias.

The setting of our experiment is clearly inadequate for collecting long series of individual data as the attention and motivation of participants already drop at the fourth time step. Designing a completely different experiment, that would provide long term individual data, is a serious challenge. Large scale long lasting game experiment on the internet, like for instance the one reported in [45], could offer new means to address this issue.

Finally, it seems noticeable that our main result, the existence of a bias from sensitivity to feedbacks, originates in theoretical agent simulations. This bias was indeed identified because its effects were easily observed in long lasting simulations, involving millions of virtual interactions. We could then detect its much smaller effect on short simulations, that we had initially overlooked. Similarly, it seems almost impossible to observe this bias in real life without looking for it with specific computations on data from a specific experiment. This is a case,

common in physics but not so much in social sciences, of an initially purely theoretical concept whose existence is confirmed experimentally.

A second case of theoretical bias could be experimentally confirmed in the near future. Indeed, the theoretical work on the agent model identifies a second bias from the decreasing sensitivity to feedbacks, a negative bias on the evaluation of others [20, 21]. This bias has also not been observed yet and designing a new experiment to detect it is another serious challenge. The experiment reported in this paper seems an interesting starting point to take up this challenge.

## Supporting information

**S1 File.**
(PDF)

**S1 Appendix. Linear mixed effect models.** The appendix explains how we computed linear mixed effect models for data structured in different levels [35] using the `lme4` R package [34].
(PDF)

**S1 Table. Sensitivity to positive or negative feedbacks.** The table shows the slopes $c_p$ and $c_n$ of the sensitivity to positive and negative feedbacks for different values of trust and different sets of time steps. Overall, the slope $c_n$ appears stronger and more significant than slope $c_p$. This suggests that the bias from sensitivity to feedbacks is mainly due to the sensitivity to negative feedbacks, especially for high trust. Moreover, for participants of high trust, $c_p - c_n$ the derivative of the self-enhancement bias is positive, suggesting that the self-enhancement bias increases with the self-evaluation. This is not true only for participants reporting low trust and $t \in (1:2)$.
(PDF)

**S2 Table. Slope of sensitivity $c$ for low ($f_0 \leq 40$) and high ($f_0 \geq 60$) anchor.** The table shows the slope of the sensitivity to feedbacks for sets distinguishing participants starting with low or high anchor and reporting different levels of trust. In sets of participants reporting high trust, the decreasing of sensitivity is significant only when the anchor is high, which confirms the pilot studies. However, in the set of participants reporting low trust, the tendency is inverted: the slope of the sensitivity is significant only when the anchor is low. These results suggest that the sensitivity to the feedbacks is not linear as it decreases more significantly when the self-evaluation is in some ranges of values, like a logistic function for instance. Moreover, the range of self-evaluation for which the sensitivity decreases more significantly depends on the level of trust and possibly on the related level of involvement or attention.
(PDF)

**S3 Table. Bias from sensitivity $S'$ for low ($f_0 \leq 40$) and high ($f_0 \geq 60$) anchor.** The table shows the bias from sensitivity for sets distinguishing participants starting with low or high anchor and reporting different levels of trust. In sets of participants reporting high trust, the bias from sensitivity is significant only when the anchor is high, which confirms the pilot studies. However, in the set of participants reporting low trust, the tendency is inverted: the bias from sensitivity is significant only when the anchor is low.
(PDF)

**S4 Table. Self-enhancement bias $E$ for $t \in (1:2)$.** The table shows the variations of the measures of self-enhancement bias $E$ computed for $t \in (1:2)$ with scale, gender and self-esteem. The main features are similar to the ones of the same table for $t \in (1:2)$ shown in the main text, with a higher standard deviation for time steps in $(1:2)$ because the sets are smaller.
(PDF)

**S5 Table. Theoretical sensitivity bias $S'$ for $t \in (1:2)$.** The table shows the variations of the measures theoretical sensitivity bias $S'$ for $t \in (1:2)$ with scale, gender and self-esteem. The main features are similar to the ones of the same table for $t \in (1:3)$ and $t \in (1:4)$, with a higher standard deviation for time steps in $(1:2)$ because the sets are smaller.
(PDF)

**S6 Table. Self-enhancement bias $E$ for $t \in (1:3)$.** The table shows the variations of the measures of self-enhancement bias $E$ computed for $t \in (1:3)$ with scale, gender and self-esteem.
(PDF)

**S7 Table. Theoretical sensitivity bias $S'$ for $t \in (1:3)$.** The table shows the variations of the measures theoretical sensitivity bias $S'$ for $t \in (1:3)$ with scale, gender and self-esteem.
(PDF)

**S8 Table. Self-enhancement bias $E$ for $t \in (1:3)$.** The table shows the variations of the measures of self-enhancement bias $E$ computed for $t \in (1:3)$ with scale, gender and self-esteem.
(PDF)

**S9 Table. Theoretical sensitivity bias $S'$ for $t \in (1:3)$.** The table shows the variations of the measures theoretical sensitivity bias $S'$ for $t \in (1:3)$ with scale, gender and self-esteem.
(PDF)

**S10 Table. Slope $c$ of sensitivity to feedback for interview time greater than 3 minutes.** The table shows the slope of the sensitivity to feedbacks, when removing the 14 participants who took less than 3 minutes to fill the questionnaire from the data (56 triples $a_t^i, \delta_t^i, a_{t+1}^i$ removed from the data).
(PDF)

**S11 Table. Bias $S'$ from sensitivity to feedbacks for interview time greater than 3 minutes.** The table shows the slope of the sensitivity to feedbacks and the mean and standard deviation of the bias from sensitivity computed on 200 bootstrap samples, when removing the 14 participants who took less than 3 minutes to fill the questionnaire from the data (56 triples $a_t^i, \delta_t^i, a_{t+1}^i$ removed from the data).
(PDF)

## Acknowledgments

We are grateful to Sylvie Huet for her support at early stages of the research.

## Author Contributions

**Conceptualization:** Guillaume Deffuant, Thibaut Roubin, Armelle Nugier, Serge Guimond.

**Formal analysis:** Guillaume Deffuant.

**Investigation:** Guillaume Deffuant.

**Methodology:** Guillaume Deffuant, Armelle Nugier, Serge Guimond.

**Software:** Thibaut Roubin.

**Writing – original draft:** Guillaume Deffuant.

**Writing – review & editing:** Guillaume Deffuant, Armelle Nugier, Serge Guimond.

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
