## [Decision Letter · Decision Letter 0]

26 Apr 2023

PONE-D-22-34148A newly detected bias in self-evaluationPLOS ONE

Dear Dr. DEFFUANT,

Thank you for submitting your manuscript to PLOS ONE. After careful consideration, we feel that it has merit but does not fully meet PLOS ONE’s publication criteria as it currently stands. Therefore, we invite you to submit a revised version of the manuscript that addresses the points raised during the review process.

In line with reviewers' comments, your major revision will require:1. More detailed theoretical background (see helpful comments by reviewer #2) and introduction, including the rationale for all included variables in your models.2. Restating of the research goals so that they correspond more closely to your analyses.3 .A different, more suitable analytical approach (multilevel modelling that aligns with the research design) 4. More clarity and detailed report on data analysis e.g., model specifications, diagnostics. Also making your code available - I suggest providing a link to a GitHub account with the code -  would help clarify some of concerns and questions of the reviewers regarding the analyses you have done. Finally, the revision will require including some additional tables and figures - possibly as Supplementary Materials (see reviewer #2 comments).Since both reviewers, as myself, recognize the potential of the paper I would like to give you a chance to resubmit with major revisions. However, I recognize that it will require considerable changes.It is my impression that the reviewers provided well-explained and helpful comments and I hope that will enable you to make your revisions.

I want to emphasize that the major revision should address each and every comment by both reviewers fully for paper to be accepted.

We look forward to receiving your revised manuscript.

Kind regards,

Srebrenka Letina, Ph.D.

Academic Editor

PLOS ONE

2. Please provide additional details regarding participant consent. In the Methods section, please ensure that you have specified (1) whether consent was informed and (2) what type you obtained (for instance, written or verbal). If your study included minors, state whether you obtained consent from parents or guardians. If the need for consent was waived by the ethics committee, please include this information.

“This research has been partly supported by the French National Research Agency through the ToRealSim project”

“We are grateful to Sylvie Huet for her support at early stages of the research.”

“This research has been partly supported by the French National Research Agency through the ToRealSim project”

Additional Editor Comments:

One reviewer suggested to reject this paper, while the other suggested a major revision. Upon closer inspection of their reasoning it seems that they both refer to similar general issues that require re-stating of the research goals, more effort in providing suitable background rationale and a rather different analytical approach. But both reviewers, as myself, recognize the potential of the paper. Therefore, if you are able to make such considerable changes in the available time, I invite to resubmit your major revision.

I want to emphasize that the major revision should address each and every comment by both reviewers fully for paper to be accepted.

Reviewers' comments:

Reviewer's Responses to Questions

**Comments to the Author**

1. Is the manuscript technically sound, and do the data support the conclusions?

Reviewer #1: No

Reviewer #2: Yes

2. Has the statistical analysis been performed appropriately and rigorously? 

Reviewer #1: I Don't Know

Reviewer #2: I Don't Know

3. Have the authors made all data underlying the findings in their manuscript fully available?

Reviewer #1: Yes

Reviewer #2: Yes

4. Is the manuscript presented in an intelligible fashion and written in standard English?

Reviewer #1: Yes

Reviewer #2: Yes

5. Review Comments to the Author

Reviewer #1: I reviewed a previous version of this paper. In this revision the authors have addressed some of the previous concerns, for example by offering a rationale for why higher self-evaluations for a highly specific task might be associated with lower responsiveness to feedback on performance. The paper’s underlying idea remains interesting, a new process (different from conventional self-enhancement) that can lead to exaggerated self-evaluations. A number of concerns remain, however. In the most general terms, the main problem is a mismatch between the paper’s stated goal, testing a theoretical prediction derived from earlier modeling work in a human-subjects experiment, and what is actually done, which has a strongly exploratory flavor. There are also important areas of unclarity, for example about the details of the analyses.

The paper’s stated goal would best be met by an analysis that (a) omitted participants who reported low levels of trust in the manipulation – because such participants are not relevant for testing the hypothesis – and (b) used all the remaining participants in a multilevel analysis that accounts for nonindependence of responses from the same individual. As far as I can tell, no such analysis is reported. Instead, the analyses mostly retain the inappropriate low-trust participants, and slice and dice the data into many subsets for separate analyses (for example, only the first two or the first 3 responses from the 4 trials; males versus females; high versus low self-esteem; two different scoring methods), as seen in Tables 4 through 10. The problem is that multiple low-powered analyses (low-powered because they use only subsets of the data) give highly variable results, and some will pop up as significant by chance alone without these results being meaningful. A better approach would be to conduct a single analysis using all high-trust participants to test the overall hypothesis, and perhaps follow that up with exploratory analyses to examine trends by gender or self-esteem, etc.

A related concern is that the potential moderator variables that are examined (scoring method, gender, self-esteem and so on) are not theoretically motivated – there is no conceptual discussion of why these variables should influence the magnitude of the effect. In fact, one atheoretical factor (rank versus score) ends up reversing the predicted effect (top of p. 24), which can be given only a speculative interpretation. The whole paper comes across as a large-scale exercise in data-fishing rather than as a focused test of an interesting a priori hypothesis.

There is still a concern that participants were required to report a new self-rating between their former rating and the new feedback – in other words, the program forced participants to modify their self-evaluations in the way described by the mathematical model. This may partially account for any agreement between the experimental data and the theoretical predictions.

The exact details of the multilevel (hierarchical) analyses HLM1 and HLM2 not specified on p. 9. The model equation or model specification in lmer syntax for R (for example) should be provided.

Reviewer #2: The authors present a very interesting paper dedicated to investigate the interplay between self-esteem, sensitivity to feedback and self-enhancement. They propose to research this important topic by an experimental study in which participants assess an area of irregular geometric shapes and then self-assess their performance. In addition, participants are given experimentally manipulated feedback, which is either positive or negative. Overall, this is a very interesting study with potentially novel contribution.

However, reading the article I could not shrug off the impression that the paper resembles more a technical report than a full-blown research paper. This is mainly due to a very cursory introduction, in which, in my opinion, the theses of the research are not sufficiently introduced and explained. The authors refer to agent-model studies that have established a relation between the level of self-assessment and sensitivity to feedback, pointing to a decreased sensitivity to feedback as one of the sources of self-enhancement. This is a very interesting research, but I would like to see more theoretical elaboration on it as simple references to some previous studies do not suffice. The literature on sources and mechanisms of self-enhancement is vast, so I recommend introducing the ideas of the research in the context of a proper literature review. The list of exemplary publications to which the authors can refer contains, i.a.:

Abele, A. E., & Wojciszke, B. (2007). Agency and communion from the perspective of self versus others. Journal of Personality and Social Psychology, 93(5), 751–763. https://doi.org/10.1037/0022-3514.93.5.751

Beer, J. S. (2014). Exaggerated Positivity in Self-Evaluation: A Social Neuroscience Approach to Reconciling the Role of Self-esteem Protection and Cognitive Bias. Social and Personality Psychology Compass, 8(10), 583–594. https://doi.org/10.1111/spc3.12133

Beer, J. S., Rigney, A. E., & Koski, J. E. (2018). Self-evaluation. In J. T. Wixted (Ed.), Stevens’ Handbook ofExperimental Psychology and Cognitive Neuroscience. Fourth Edition. (pp. 1–30). John Wiley & Sons. https://doi.org/10.7551/mitpress/7458.003.0023

Cahill, D. P. (2015). Wishful Thinking, Fast and Slow. Doctoral dissertation, Harvard University, Graduate School of Arts & Sciences. https://dash.harvard.edu/bitstream/handle/1/17467495/CAHILL-DISSERTATION-2015.pdf?sequence=1

Gesiarz, F., Cahill, D., & Sharot, T. (2019). Evidence accumulation is biased by motivation: A computational account. PLoS Computational Biology, 15(6), 1–15. https://doi.org/10.1371/journal.pcbi.1007089

Miller, T. M., & Geraci, L. (2011). Training metacognition in the classroom: The influence of incentives and feedback on exam predictions. Metacognition and Learning, 6(3), 303–314. https://doi.org/10.1007/s11409-011-9083-7

Paulhus, D. L., & Reid, D. B. (1991). Enhancement and Denial in Socially Desirable Responding. Journal of Personality and Social Psychology, 60(2), 307–317. https://doi.org/10.1037/0022-3514.60.2.307

Rigney, A. E. (2019). The role of biased searching through memory in motivated social evaluation. Unpublished doctoral dissertation. University of Texas, TX, USA.

Sedikides, C., & Green, J. D. (2000). On the self-protective nature of inconsistency-negativity management: Using the person memory paradigm to examine self-referent memory. Journal of Personality and Social Psychology, 79(6), 906–922. https://doi.org/10.1037/0022-3514.79.6.906

Sedikides, C., & Gregg, A. P. (2008). Portraits of the self. The SAGE Handbook of Social Psychology, 93–122. https://doi.org/10.4135/9781848608221.n5

Swann, W. B., & Read, S. J. (1981b). Acquiring self-knowledge: The search for feedback that fits. Journal of Personality and Social Psychology, 41(6), 1119.

Trope, Y., & Neter, E. (1994). Reconciling competing motives in self-evaluation: the role of self-control in feedback seeking. Journal of Personality and Social Psychology, 66(4), 646- 657. https://doi.org/10.1037/0022-3514.66.4.646

The authors should also comment on the role of the domain in which participants self-assess and receive feedback with regard to its relevance/importance and agency/communion values.

Besides the request to write a much broader literature review, I would also like to rise some minor issues:

Anchor: Why 1/3 of the participants were given low anchor and 2/3 high anchor and not 50:50? Was that done solely to have more data for respondents with higher anchor?

Research design: In general, authors should re-write that part of the paper and explain their experimental design in clear and short terms. If I am not mistaken, the design is in fact close to 6 (type of feedback sequence) x 2 (type of feedback: score or rank), so authors should present it more clearly in terms understandable to interdisciplinary readership. I also think that the paper’s clarity would benefit from commenting and explaining more about the regression equations used to analyse the results, including their hierarchical character. I think some graphs or diagrams could help here, otherwise the reader is a bit lost in the overflow of equations and tables.

Slopes and Intercepts: I would appreciate if the authors could possibly present all the regression coefficients in a clear way before presenting the numerical results. Otherwise, it is difficult to follow what all those different slopes and intercepts mean.

Tables and Figures layout: The tables could be a bit more readable and more conforming to standards of communication, e.g., explaining what does the “*” stand for (I assume it’s for the p-levels but it is not explained in the paper). Also, the tables and figures lack titles.

Self-disparagement: I would also suggest replacing the term “self-disparagement” with something more often used in the self-motives literature, e.g. terms “self-effacement”, “self-derogation”, or “self-diminishment” (e.g. Kiu, Chiu & Zou, 2010). “Self-disparagement” seems to be used in other contexts, e.g. depression or humour research. In my opinion, we should avoid multiplying terms and should stick to the already many terms used in the self-motives field.

Gender differences: I think the gender differences in the self-enhancement bias described in the paper are not entirely new and the authors should comment on them in the light of literature. For example, the authors can use the works on math-related self-enhancement of:

Paulhus, D. L., & John, O. P. (1998). Egoistic and moralistic biases in self‐perception: The interplay of self‐deceptive styles with basic traits and motives. Journal of personality, 66(6), 1025-1060.

Palczyńska, M., & Rynko, M. (2021). ICT skills measurement in social surveys: Can we trust self-reports?. Quality & Quantity, 55(3), 917-943.

In general, males have higher self-enhancement tendencies in agentic than communal domains. Please link your findings with literature on this topic.

Self-esteem and self-enhancement: The authors should also elaborate more on this topic in the discussion section, e.g. starting from the discussion in:

Gebauer, J. E., Wagner, J., Sedikides, C., & Neberich, W. (2013). Agency-communion and self-esteem relations are moderated by culture, religiosity, age, and sex: Evidence for the “self-centrality breeds self-enhancement” principle. Journal of Personality, 81(3), 261–275. https://doi.org/10.1111/j.1467-6494.2012.00807.x

Kernis, M. H. (2003). Toward a conceptualization of optimal self-esteem. Psychological Inquiry, 14(1), 1-26.

Screening out participants: I would like to see some justification on using a broad RT filter based on 3 minutes. Why this value has been chosen? The literature on using response times in screening out careless respondents advocates for a different, more theory-based approach. The authors should present the distribution of response times and base their further moves on previous works, e.g.:

Huang, J. L., Curran, P. G., Keeney, J., Poposki, E. M., & DeShon, R. P. (2012). Detecting and deterring insufficient effort responding to surveys. Journal of Business and Psychology, 27, 99-114.

Kroehne, U., Buchholz, J., & Goldhammer, F. (2019). Detecting carelessly invalid responses in item sets using item-level response times. In Annual Meeting of the National Council on Measurement in Education.

Ulitzsch, E., Pohl, S., Khorramdel, L., Kroehne, U., & von Davier, M. (2022). A response-time-based latent response mixture model for identifying and modeling careless and insufficient effort responding in survey data. psychometrika, 87(2), 593-619.

Consequences of screening: Basing on the consensus in careless responding research when any participants’ screening out is proposed, its consequences should be clearly showed, i.e., all analyses should be conducted both on full sample and on screened out (filtered) sample. Please present such results.

Trust: Why the threshold of trust higher or equal to 7 was chosen? Would the results differ if other cut-off point have been picked up?

From very minor issues – the paper is in general well-written, but there are occasional typos and similar minor linguistic problems so I recommend proofreading before final submission.

6. PLOS authors have the option to publish the peer review history of their article (what does this mean?). If published, this will include your full peer review and any attached files.

Reviewer #1: No

Reviewer #2: **Yes: **Marek Muszyński

---

## [Author Response · Author response to Decision Letter 0]

15 Jun 2023

Our response to the reviewers is detailed in the cover letter to the editor.

---

## [Decision Letter · Decision Letter 1]

17 Jul 2023

PONE-D-22-34148R1A newly detected bias in self-evaluationPLOS ONE

Dear Dr. DEFFUANT,

Thank you for submitting your manuscript to PLOS ONE. After careful consideration, we feel that it has merit but does not fully meet PLOS ONE’s publication criteria as it currently stands. Therefore, we invite you to submit a revised version of the manuscript that addresses the points raised during the review process.

Although reviewers recognized the upgraded quality of the revised manuscript and generally show an interest and a belief in its contributions, a major revision is still required in order for the manuscript be published. Both reviewers still require a list of improvements. Some of the main points include:- Justify the choice of responses provided to participants and clearly state the consequences of such choice for your findings- Justify/explain the sample size and undertake a power analysis- Theoretically justify all of your analysis and examine the findings accordingly-  Rethink your statistical approach in general, specifically consider using hierarchical modelling approach, if not applicable explain in details why and what are the limitations of not doing so- Strengthen the Introduction theoretically (with works suggested by reviewers), but also the Discussion, including sections on future research and limitations- Make a more substantial effort in providing figures that help in understanding the paper's findingsThese are just main improvements expected in the next revision, but each point made by both reviewers (see the details further below) needs to be addressed fully and convincingly for the paper to be accepted.I hope you will find the reviewer's comments helpful for your next revision.

We look forward to receiving your revised manuscript.

Kind regards,

Srebrenka Letina, Ph.D.

Academic Editor

PLOS ONE

Additional Editor Comments (if provided):

Major revision required.

Reviewers' comments:

Reviewer's Responses to Questions

**Comments to the Author**

1. If the authors have adequately addressed your comments raised in a previous round of review and you feel that this manuscript is now acceptable for publication, you may indicate that here to bypass the “Comments to the Author” section, enter your conflict of interest statement in the “Confidential to Editor” section, and submit your "Accept" recommendation.

Reviewer #1: (No Response)

Reviewer #2: All comments have been addressed

2. Is the manuscript technically sound, and do the data support the conclusions?

Reviewer #1: Yes

Reviewer #2: Yes

3. Has the statistical analysis been performed appropriately and rigorously? 

Reviewer #1: I Don't Know

Reviewer #2: Yes

4. Have the authors made all data underlying the findings in their manuscript fully available?

Reviewer #1: Yes

Reviewer #2: Yes

5. Is the manuscript presented in an intelligible fashion and written in standard English?

Reviewer #1: Yes

Reviewer #2: Yes

6. Review Comments to the Author

Reviewer #1: This relatively thorough revision has improved the paper in several respects. I still have reservations which I will list briefly below, but the paper now seems to offer enough value to warrant publication. The empirical demonstration of a novel process leading to self-enhancement, evaluation-dependent sensitivity to feedback, is a solid contribution.

The authors may or may not wish to address my remaining concerns.

1. The experiment forced participants to respond as theory specifies (making each self-assessment update fall between the previous self-assessment and the current feedback). The paper notes that in pilot testing, a majority of participants responded in this way without being restricted. However, the restriction still troubles me – it shapes the way participants think about the task, clearly pushing them in the direction of conforming to the theoretical prediction.

2. The primary hypotheses (line 69) are solid and well-motivated. In contrast, the secondary objective is to examine several potential moderator variables which are not introduced with any conceptual rationale. This is especially troubling for the evaluation scale, which has strong and unpredicted effects on the results. One previous study found a somewhat similar pattern, but no compelling theoretical interpretation can be offered in this paper (lines 490-518).

3. I still do not think the data analysis strategy is optimal. First, participants who express low “trust” in the experimental deception are irrelevant to the hypotheses and should not be included in the analysis. Second, the analyses in the main manuscript inappropriately use multiple observations ("triples") from each participant, which are statistically non-independent in violation of the assumptions of ordinary regression analysis. The analyses in SM use a two-step approach, running a regression for each participant and then predicting the coefficients of those analyses as outcome measures. This approach suffers from low power, because in the first step each analysis has very few observations (and so few coefficients are significant, line 18 of SM). More appropriate would be a hierarchical linear analysis, for example with the lmer function in R. Such an analysis does not use two separate steps but properly accounts for the non-independence of multiple observations from each participant, and gives results that describe the entire sample.

As stated above, however, despite these conceerns I think the paper offers meaningful value to the field.

Reviewer #2: I thank the authors for addressing my comments.

I think that the present version of the article is markedly better than the previous one. Well done!

However, I still think that some parts of the manuscript should be improved before publication. I present my concerns below:

1. Please provide power analysis to justify that the sample sizes enable achieving required statistical power in your analyses.

2. The introduction still needs improving. I think the authors should clearly introduce topics such as agency and communion (with relation to self-enhancement) or different types of self-enhancement (impression management, self-deceptive denial, self-deceptive enhancement). Please elaborate more on that in the introduction, e.g., by using many excellent works of Delroy Paulhus (types of self-enhancement) or Bogdan Wojciszke (agency and communion). Please also refer to this recent article that seems relevant and also comments on inter-gender differences:

Lalwani, A. K., Lee, H., Shrum, L. J., & Viswanathan, M. (2023). Men engage in self‐deceptive enhancement, whereas women engage in impression management. Psychology & Marketing.

3. Please also include more information about the relations between self-esteem, gender and perceived level of expertise with self-enhancement in the introduction. An excellent paper by Stav Atir can help you to achieve this:

Atir, S., Rosenzweig, E., & Dunning, D. (2015). When knowledge knows no bounds: Self-perceived expertise predicts claims of impossible knowledge. Psychological Science, 26(8), 1295-1303.

4. Please, please think hard on how can you improve the presentation of your findings in a graphical way. For now, it is hard to follow the information flow presented in Tables 4-7. I would also like to point your attention to the fact that they do not conform to any widely accepted style. I would suggest to rethink their "look" and redraw them in the APA style. This also relates to the * for p-levels. Still, I do not know to which number of stars is related to which p-level.

5. I was also expecting some results about the role of gender and self-esteem play in feedback sensitivity and self-enhancement presented more in a fashion of regresssion results. Could you possibly present it in this way to increase understanding of the analyses and results?

6. In lines 75-76 and 187 you use the term "modulated". I think you should use "moderated" or "related to", depending on what is your analytic stance towards those relations. "Modulated" is a conversational term and should not be used here (I know it is used widely, but I think we should avoid it).

7. Please tone down the language in lines 58-63. These are only hypotheses/suggestions.

8. Regarding lines 100-107 - can you provide any evidence from studies with human participants?

9. Short question about Table 1 - why does it contain only four combinations? There are more combinations of balanced six feedbacks...

10. It is outstanding that you have found such a valuable and relevant references as number 19 (Muller-Pinzler et al., 2019)! I think you should capitalise more on that and present this important study more in the introduction. Please, do not assume that the readership knows this paper. It should be sufficiently presented and commented on in the relevant sections of the paper.

11. Please expand the discussion on self-esteem and self-enhancement on topics such as sensitivity to feedback, narcissism, fragile self-esteem and rely to relevant literature.

12. The paper lacks important sections such as "Limitations" and "Future directions". I would ask authors to amend for that.

I see that the paper is approaching the right direction and would like to encourage the authors for some more effort before the publication. Fingers crossed!

7. PLOS authors have the option to publish the peer review history of their article (what does this mean?). If published, this will include your full peer review and any attached files.

Reviewer #1: No

Reviewer #2: No

---

## [Author Response · Author response to Decision Letter 1]

25 Aug 2023

The complete response to the reviewers is in the file "answerToTheRevewers_R3.pdf".

---

## [Decision Letter · Decision Letter 2]

8 Oct 2023

PONE-D-22-34148R2A newly detected bias in self-evaluationPLOS ONE

Dear Dr. DEFFUANT,

Thank you for submitting your manuscript to PLOS ONE. After careful consideration, we feel that it has merit but does not fully meet PLOS ONE’s publication criteria as it currently stands. Therefore, we invite you to submit a revised version of the manuscript that addresses the points raised during the review process. A major revision is required. In the next revision it is expected that you will address the review #1 suggestions appropriately - namely, by using hierarchical multi-level modeling. Also, address the issue related with artificially imposed restriction by recognizing it explicitly in the section about limitation of the study.  Finally, we require that you address all minor comments made by reviewer #2 (see below).

We look forward to receiving your revised manuscript.

Kind regards,

Srebrenka Letina, Ph.D.

Academic Editor

PLOS ONE

Additional Editor Comments (if provided):

Major revision required.

Reviewers' comments:

Reviewer's Responses to Questions

**Comments to the Author**

1. If the authors have adequately addressed your comments raised in a previous round of review and you feel that this manuscript is now acceptable for publication, you may indicate that here to bypass the “Comments to the Author” section, enter your conflict of interest statement in the “Confidential to Editor” section, and submit your "Accept" recommendation.

Reviewer #1: (No Response)

Reviewer #2: All comments have been addressed

2. Is the manuscript technically sound, and do the data support the conclusions?

Reviewer #1: Partly

Reviewer #2: Yes

3. Has the statistical analysis been performed appropriately and rigorously? 

Reviewer #1: No

Reviewer #2: I Don't Know

4. Have the authors made all data underlying the findings in their manuscript fully available?

Reviewer #1: Yes

Reviewer #2: Yes

5. Is the manuscript presented in an intelligible fashion and written in standard English?

Reviewer #1: Yes

Reviewer #2: Yes

6. Review Comments to the Author

Reviewer #1: This second revision is very similar in substance to the previous version. The theoretical point that differential sensitivity to feedback is a novel mechanism that can contribute to self-enhancement is sound. But I still have two significant concerns about the experimental demonstration reported in this paper.

First, as noted before: “The experiment forced participants to respond as theory specifies (making each self-assessment update fall between the previous self-assessment and the current feedback). The paper notes that in pilot testing, a majority of participants responded in this way without being restricted. However, the restriction still troubles me – it shapes the way participants think about the task, clearly pushing them in the direction of confirming the theoretical prediction.” The authors’ response emphasizes that this restriction was intended to reduce noise in the observations, and that participants remained free to choose any self-evaluation between the given limits. But that response misses my point that the restriction imposes on participants a specific way of thinking about the task. By forcing them to think of it as updating their previous response by moving some proportion of the way toward the current feedback, the researchers are in effect forcing them to use the theoretically assumed process. There are potentially other ways participants might approach the task if this restriction was not artificially imposed, for example deciding that the feedback jumps around too much to be useful or believable and therefore trying to disregard it. The authors’ response completely ignores this concern, that the restriction shapes the way participants think about the task.

Second, I remain concerned that the main analyses use ordinary linear regression on a set of “triples,” which include multiple observations from each participant. Those are obviously non-independent, violating a statistical assumption of the regression model. I recommended using a hierarchical regression (also termed multilevel or mixed model), in which participants are explicitly treated as a random effect, to correctly account for this non-independence.

The authors respond in two ways. One part of the authors’ response is to present regression analyses that include additional predictors (age, gender, trust, etc.). These are entirely irrelevant to the criticism I was making.

Second, they report “hierarchical” models in SM (2.1 and 2.2) that they argue show any non-independence can be ignored. These models are analyzed in two separate steps (between and within participants) rather than as a standard hierarchical model, for example using lmer* in R. I do not understand how or why the authors claim that the results from these models rule out non-independence. More important, experts recommend the use of hierarchical models in all cases when the data has a multilevel structure (here, observations nested within participants) regardless of the statistical level of non-independence. For example, Nezlek (2008)* writes “analysts should use multilevel modeling when they have a multilevel data structure – pure and simple. When I am asked for advice regarding whether or not multilevel modeling is appropriate, my first question concerns the nature of the data structure. If there is a meaningful nested hierarchy to the data, my advice is to use multilevel modeling, irrespective of distracting arguments about ICCs and so forth.” And “researchers sometimes use a low or zero ICC to justify a decision not to use multilevel modeling – on the grounds that because there is no (or very little) between-group variance in the dependent measure, the grouped (or nested) structure of the data can be ignored. This is a dangerous assumption that is not justifiable. Frequently (or almost invariably), researchers are interested in relationships between measures. The fact that there is little or no between-group variance in a measure does not mean that the relationship between this measure and another measure is the same across all groups, something that is assumed if one conducts and analysis that ignores the grouped structure of the data. By extension, even if there is no between-group variance for all of the measures of interest, it cannot be assumed that relationships between or among these measures do not vary across groups.”

To be absolutely clear, the model I view as appropriate, expressed as an lmer* formula, is something like:

Dependent.variable ~ 1 + independent.variables + (1 + independent.variables | Participant)

where the term in parentheses indicates random slopes and intercepts per participant.

*References

Nezlek, J. B. (2008). An introduction to multilevel modeling for social and personality psychology. Social and Personality Psychology Compass, 2(2), 842-860.

https://cran.r-project.org/web//packages/lme4/vignettes/lmer.pdf

Reviewer #2: I think that the manuscript is improved but it is still written in a unnecessarily complicated way, even convoluted sometimes. The literature review is still shallow at times, although the changes introduced in the review rounds have increased the quality and comprehensibility of the paper.

I feel that I cannot provide more suggestions for the authors and it is up to the editors to decide on further fate of the manuscript. I just have few remaining suggestions/questions (the number is the number of line in the corrected manuscript):

a) 28-29 "in its various dimensions"

I think the paper relates to self-enhancement in the sense of self-exaggeration?

b) 109-112

Is it really completely different from other biases?

Please tone down the language here as the originality of the effects is not as high as you claim.

Take a look at feedback sensitivity research available, e.g. this article by Filip Gęsiarz:

Gesiarz, F., Cahill, D., & Sharot, T. (2019). Evidence accumulation is biased by motivation: A computational account. PLOS Computational Biology, 15(6), e1007089.

This topic was also researched by Stav Atir or Jennifer S. Beer, for example.

c) 303

Table 1 says "six series of 4 feedbacks" but I rather see four series of six different feedbacks?

d) 476

you do not have sample size of 6000, you just have 1500 that you use in 4, if I am right, different analyses, n'est-ce pas?

e) 553

I think that treating continuous variables as dichotomous is very questionable, I advice to redo the analyses presented in Figure 6 with trust and self-esteem treated as continuous predictors

f) 583

You quite often refer to differences as "more significant" or "very significant". How about using effect size measures to quantify such differences? Please provide effect size measures for your key effects.

g) Citations - I suggest to take APA style but it is up to the editors to decide on that.

h) spelling mistakes:

Table 1. heading contains a small mistake ("tow" instead of "two")

7. PLOS authors have the option to publish the peer review history of their article (what does this mean?). If published, this will include your full peer review and any attached files.

Reviewer #1: No

Reviewer #2: No

---

## [Author Response · Author response to Decision Letter 2]

27 Oct 2023

The response to the reviewer is provided in a specific file submitted via the system (answerToReviewerR4.pdf)

---

## [Decision Letter · Decision Letter 3]

24 Nov 2023

PONE-D-22-34148R3A newly detected bias in self-evaluationPLOS ONE

Dear Dr. DEFFUANT,

Thank you for submitting your manuscript to PLOS ONE. After careful consideration, we feel that it has merit but does not fully meet PLOS ONE’s publication criteria as it currently stands. Therefore, we invite you to submit a revised version of the manuscript that addresses the points raised during the review process.

For paper to be accepted you are required to:1. Address the remaining comments of reviewer #2 (see below)2. Address previous comments made by reviewer #1 (see previous reviews) that in my opinion have not been addressed in detail and in convincing manner yet.

We look forward to receiving your revised manuscript.

Kind regards,

Srebrenka Letina, Ph.D.

Academic Editor

PLOS ONE

Journal Requirements:

Reviewers' comments:

Reviewer's Responses to Questions

**Comments to the Author**

1. If the authors have adequately addressed your comments raised in a previous round of review and you feel that this manuscript is now acceptable for publication, you may indicate that here to bypass the “Comments to the Author” section, enter your conflict of interest statement in the “Confidential to Editor” section, and submit your "Accept" recommendation.

Reviewer #2: All comments have been addressed

2. Is the manuscript technically sound, and do the data support the conclusions?

Reviewer #2: Yes

3. Has the statistical analysis been performed appropriately and rigorously? 

Reviewer #2: Yes

4. Have the authors made all data underlying the findings in their manuscript fully available?

Reviewer #2: Yes

5. Is the manuscript presented in an intelligible fashion and written in standard English?

Reviewer #2: Yes

6. Review Comments to the Author

Reviewer #2: I would like to thank the authors for all the reactions and comments to my suggestions. In general, I think that the responses satisfy my curiosity and adequately address all the caveats I had.

Some comments that remain:

a) 109-112:

I think that your discussion of the similarities and differences between your paper and the works of Gesiarz et al. (2019) and Atir et al. (2015) is very interesting and you should include it somewhere in the paper (maybe in the Discussion section?). One of the main tasks of the authors of any paper, especially one that claims to have found something new, is to comprehensively compare and contrast new findings with what was previously done in the field. It does not suffice to say “it’s new”, you have to show it. Sometimes it is really new and there are very few studies that directly links to your findings. Then you have to dig somewhat deeper and bring forth also the studies with more indirect links to what you have done.

In my opinion, your discussion does not discuss the many studies indirectly related to your experiments, therefore, I would ask you to improve it by addressing other studies about the interplay of feedback, self-enhancement, self-esteem, etc. This discussion should include studies like Gesiarz et al. (2019) as the task here is also to contrast the present findings with the previous ones and to show, where new knowledge was provided.

Some other studies you may find useful:

Haddara, N., & Rahnev, D. (2022). The impact of feedback on perceptual decision-making and metacognition: Reduction in bias but no change in sensitivity. Psychological Science, 33(2), 259-275.

Heck, P. R., & Krueger, J. I. (2015). Self-enhancement diminished. Journal of Experimental Psychology: General, 144(5), 1003–1020. https://doi.org/10.1037/xge0000105 (Study 3)

Jussim, L., Yen, H., & Aiello, J. R. (1995). Self-consistency, self-enhancement, and accuracy in reactions to feedback. Journal of experimental social psychology, 31(4), 322-356.

Schulz, L., Fleming, S. M., & Dayan, P. (2023). Metacognitive computations for information search: Confidence in control. Psychological Review.

I would also like to remind the authors that it is their task to discuss their findings in such a manner to convince the reviewers that the discussion is comprehensive and well-based in the extant literature.

b) 303

OK, now I see what you see and we see it eye to eye. Maybe adding a first column “Serie” or something like that would enhance this table further still, but now I think it is ok. Thank you.

c) 476

OK, this is rather a semantic discussion – “sample size” is typically used to indicate number of observations, which, in this case, can be either participants or answers. My recommendation is to reframe this section to make it clear what you count as “sample size” in each of the instances/analyses to avoid any confusion between the number of participants and the number of answers collected from them. I also think that the power calculation should be re-run now, to accommodate to the new analytic model employed (hierarchical model with mixed effects).

d) 553

OK, so be it, please consider adding this point somewhere to the Limitations or Future Directions section then.

Otherwise, I have no further comments and I think that this article would be ready for publication after adequate addressing of the present recommendations.

7. PLOS authors have the option to publish the peer review history of their article (what does this mean?). If published, this will include your full peer review and any attached files.

Reviewer #2: No

---

## [Author Response · Author response to Decision Letter 3]

3 Dec 2023

The response to the reviewers is in the file "answerToReviewersR5.pdf"

---

## [Editor Report · Decision Letter 4]

12 Dec 2023

A newly detected bias in self-evaluation

PONE-D-22-34148R4

Dear Dr. DEFFUANT,

We’re pleased to inform you that your manuscript has been judged scientifically suitable for publication and will be formally accepted for publication once it meets all outstanding technical requirements.

Kind regards,

Srebrenka Letina, Ph.D.

Academic Editor

PLOS ONE
---

## [Editor Report · Acceptance letter]

24 Jan 2024

PONE-D-22-34148R4 

PLOS ONE

Dear Dr. Deffuant, 

I'm pleased to inform you that your manuscript has been deemed suitable for publication in PLOS ONE. Congratulations! Your manuscript is now being handed over to our production team.

Kind regards, 

on behalf of

Dr. Srebrenka Letina 

Academic Editor

PLOS ONE